# GRAPHPATCHER: Mitigating Degree Bias for Graph Neural Networks via Test-time Augmentation

**Mingxuan Ju[1], Tong Zhao[2], Wenhao Yu[1], Neil Shah[2], Yanfang Ye[1]**
[1]University of Notre Dame, [2]Snap Inc.
[1]{mju2,wyu1,yye7}@nd.edu; [2]{tzhao,nshah}@snap.com

## Abstract

Recent studies have shown that graph neural networks (GNNs) exhibit strong biases towards the node degree: they usually perform satisfactorily on high-degree nodes with rich neighbor information but struggle with low-degree nodes. Existing works tackle this problem by deriving either designated GNN architectures or training strategies specifically for low-degree nodes. Though effective, these approaches unintentionally create an artificial out-of-distribution scenario, where models mainly or even only observe low-degree nodes during the training, leading to a downgraded performance for high-degree nodes that GNNs originally perform well at. In light of this, we propose a test-time augmentation framework, namely GRAPHPATCHER, to enhance test-time generalization of any GNNs on low-degree nodes. Specifically, GRAPHPATCHER iteratively generates virtual nodes to patch artificially created low-degree nodes via corruptions, aiming at progressively reconstructing target GNN's predictions over a sequence of increasingly corrupted nodes. Through this scheme, GRAPHPATCHER not only learns how to enhance low-degree nodes (when the neighborhoods are heavily corrupted) but also preserves the original superior performance of GNNs on high-degree nodes (when lightly corrupted). Additionally, GRAPHPATCHER is model-agnostic and can also mitigate the degree bias for either self-supervised or supervised GNNs. Comprehensive experiments are conducted over seven benchmark datasets and GRAPHPATCHER consistently enhances common GNNs' overall performance by up to 3.6% and low-degree performance by up to 6.5%, significantly outperforming state-of-the-art baselines. The source code is publicly available at https://github.com/jumxglhf/GraphPatcher.

## 1 Introduction

Graph Neural Networks (GNNs) have gained significant popularity as a powerful approach for learning representations of graphs, achieving state-of-the-art performance on various predictive tasks, such as node classification [22, 38, 9], link prediction [50, 53], and graph classification [43, 47, 11]. These tasks further form the archetypes of many real-world applications, such as recommendation systems [45, 3], predicative user behavior models [31, 52], and molecular property prediction [51, 48].

While existing GNNs are highly proficient at capturing information from rich neighborhoods (i.e., high-degree nodes), recent studies [13, 25, 36, 55] have revealed a significant performance degradation of GNNs when dealing with nodes that have sparse neighborhoods (i.e., low-degree nodes). This observation can be attributed to the fact that GNNs make predictions based on the distribution of node neighborhoods [27]. According to this line of theory, GNNs struggle with low-degree nodes due to the limited amount of available neighborhood information, which may not be able to precisely depict the learned distributions. Empirically, as shown in Figure 1, the classification accuracy of GCN [22] proportionally decays as the node degree decreases, resulting in a performance gap of ~20% accuracy. Furthermore, the sub-optimal performance of GNNs on low-degree nodes can be

37th Conference on Neural Information Processing Systems (NeurIPS 2023).

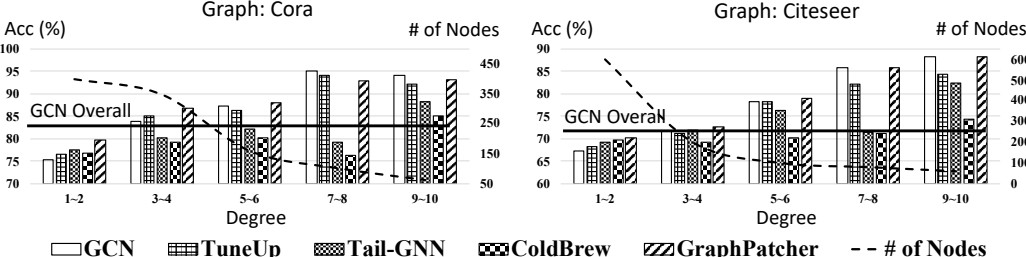

Figure 1: The classification accuracy of GCN and SoTA frameworks that mitigate degree biases.

aggravated by the power-law degree distribution commonly observed in real-world graphs, where the number of low-degree nodes significantly exceeds that of high-degree nodes [36].

To bridge this gap, several frameworks have been proposed to specifically improve GNNs' performance on low-degree nodes [36, 25, 13, 55, 49]. These frameworks either introduce designated architectures or training strategies specifically for low-degree nodes. For examples, Tail-GNN [25] enhances latent representations of low-degree nodes by incorporating high-degree structural information; whereas Coldbrew [55] retrieves a set of existing nodes as virtual neighbors for low-degree nodes. However, these approaches suffer from two significant drawbacks. Firstly, while benefiting low-degree nodes, they inadvertently create an artificial out-of-distribution scenario during training [42], where models primarily observe low-degree nodes, leading to a downgraded performance for high-degree nodes that GNNs originally perform well on. Secondly, deploying these frameworks often requires changing model architectures, which can be impractical in real-world scenarios where the original models are well-trained due to the expensive re-training cost (on large-scale graphs) and the shared usage of it across different functionalities in production.

In light of these drawbacks, we propose a test-time augmentation framework for GNNs, namely GRAPHPATCHER. Given a well-trained GNN, GRAPHPATCHER mitigates the degree bias by patching corrupted ego-graphs with multiple generated virtual neighbors. Notably, GRAPHPATCHER not only enhances the performance of low-degree nodes but also maintains (sometimes improves) GNNs performance on high-degree nodes. This behavior is empirically important because practitioners can universally apply GRAPHPATCHER to all nodes without, like previous works, manually discovering a degree threshold that differentiates the low- and high-degree nodes. To achieve so, we first generate a sequence of ego-graphs corrupted with increasing strengths. Then, GRAPHPATCHER recursively generates multiple virtual nodes to patch the mostly corrupted graph, such that the frozen GNN gives similar predictions for the patched graph and the corresponding corrupted ego-graph in the sequence. Through this scheme, GRAPHPATCHER not only learns how to patch low-degree nodes (i.e., heavily corrupted) but also maintains GNNs original superior performance on high-degree nodes (i.e., lightly corrupted). As a test-time augmentation framework, GRAPHPATCHER is parameterized in parallel with the target GNN. Hence, GRAPHPATCHER is model-agnostic and requires no updates on the target GNN, enabling practitioners to easily utilize it as a plug-and-play module to existing well-established infrastructures. Overall, our contributions are summarized as:

- We study a more practical setting of degree biases on graphs, where both the performances on low- and high-degree nodes are considered. In this case, a good framework is required to not only improve the performance over low-degree nodes but also maintain the original superior performance over high-degree nodes. We evaluate existing frameworks in this setting and observe that many of them trade off performance on high-degree nodes for that on low-degree nodes.

- To mitigate degree biases, we propose GRAPHPATCHER, a novel test-time augmentation framework for graphs. Given a well-trained GNN, GRAPHPATCHER iteratively generates multiple virtual nodes and uses them to patch the original ego-graphs. These patched ego-graphs not only improve GNNs' performance on low-degree nodes but also maintains that over high-degree nodes. Moreover, GRAPHPATCHER is applied at the testing time for GNNs, a plug-and-play module that is easily applicable to existing well-established infrastructures.

- We conduct extensive evaluation of GRAPHPATCHER along with six state-of-the-art frameworks that mitigate degree biases on seven benchmark datasets. GRAPHPATCHER consistently enhances the overall performance by up to 3.6% and low-degree performance by up to 6.5% of multiple GNNs, significantly outperforming state-of-the-art baselines.

## 2 Related Works

**Graph Neural Networks**. Graph Neural Networks (GNNs) have become one of the most popular paradigms for learning representations over graphs [22, 38, 9, 43, 23, 17, 4]. GNNs aim at mapping the input nodes into low-dimensional vectors, which can be further utilized to conduct either graph-level or node-level tasks. Most GNNs explore a layer-wise message passing scheme, where a node iteratively extracts information from its first-order neighbors, and information from multi-hop neighbors can be captured by stacked layers. They achieved state-of-the-art performance on various tasks, such as node classification [22, 44, 12, 35], link prediction [50, 53, 8], node clustering [2, 37], etc. These tasks further form the archetypes of many real-world applications, such as recommendation systems [45, 3], predictive user behavior models [31, 52], question answering [18], and molecular property prediction [51, 48, 7, 24].

**Degree Bias underlying GNNs**. Recent studies have shown that GNNs exhibit strong biases towards the node degree: they usually perform satisfactorily over high-degree nodes with rich neighbor information but suffer over low-degree nodes [13, 25, 36, 55]. Existing frameworks that mitigate degree biases derive either designated architectures or training strategies specifically for low-degree nodes. For instance, Tail-GNN [25] enhances low-degree nodes' latent representations by injecting high-degree structural information learned from high-degree nodes; Coldbrew [55] retrieves a set of existing nodes as virtual neighbors for low-degree nodes; TuneUp [13] fine-tunes the well-trained GNNs with pseudo labels and heavily corrupted graphs. Though effective for low-degree nodes, they unintentionally create an artificial out-of-distribution scenario [42], where models only observe low-degree nodes during the training, leading to downgraded performance for high-degree nodes that GNNs originally perform well at.

**Test-time Augmentation**. While data augmentations during the training phase have become one of the essential ingredients for training machine learning models [54], the augmentation applied during the testing time is far less studied, especially for the graph learning community. It has been moderately researched in the computer vision field, aimed at improving performance or mitigating uncertainties [34, 21, 41, 1]. They usually corrupt the same sample by different augmentation approaches and aggregate the model's predictions on all corrupted samples. Whereas in the graph community, GTrans [15] proposes a test-time enhancement framework, where the node feature and graph topology are modified at the test time to mitigate potential out-of-distribution scenarios.

## 3 Methodology

### 3.1 Preliminary

In this work, we specifically focus on the node classification task. Let $G = (V, E)$ denote a graph, where $V$ is the set of $|V| = N$ nodes and $E \subseteq V \times V$ is the set of $|E|$ edges between nodes. $\mathbf{X} \in \mathbb{R}^{N \times d}$ represents the feature matrix, with $i$-th row representing node $v_i$'s $d$-dimensional feature vector. $\mathbf{Y} \subseteq \{0, 1\}^{N \times C}$ denotes the label matrix, where $C$ is the number of total classes. And $\mathbf{Y}^{(L)}$ denotes the label matrix for training nodes. We denote the ego-graph of node $v_i$ is defined as $\mathcal{G}(v_i) = (V_i, E_i)$ with $V_i = \mathcal{N}_k(v_i)$, where $\mathcal{N}_k(v_i)$ stands for all nodes within the $k$-hop neighborhood of $v_i$ including itself and $E_i$ refers to the edges in-between $\mathcal{N}_k(v_i)$. A well-trained GNN $f_g(\cdot; \boldsymbol{\theta}) : G \to \mathbb{R}^{N \times C}$ parameterized by $\boldsymbol{\theta}$ takes $G$ as input and maps every node in $G$ to a $C$-dimensional class distribution. Formally, we define test-time node patching as the following:

**Definition 1** (Test-time Node Patching). *Given a GNN $f_g(\cdot; \boldsymbol{\theta})$ and a graph $G$, a test-time node patching framework $f(\cdot; \boldsymbol{\phi}) : G \to G$ takes $G$ and outputs the patched graph $\hat{G}$ with generated nodes and edges, such that the performance of $f_g$ over nodes in $G$ is enhanced when $\hat{G}$ is utilized:*

$$\arg\min_{\boldsymbol{\phi}} \ \mathcal{L}\Big(f_g\big(f(G; \boldsymbol{\phi}); \boldsymbol{\theta}^*\big), \mathbf{Y}\Big), \quad where \quad \boldsymbol{\theta}^* = \arg\min_{\boldsymbol{\theta}} \mathcal{L}\big(f_g(G; \boldsymbol{\theta}), \mathbf{Y}^{(L)}\big), \quad (1)$$

*where $\mathcal{L}$ refers to the loss function evaluating the GNN (e.g., cross-entropy or accuracy).*

In this work, we aim at mitigating the degree bias via test-time node patching. To achieve so, two challenges need to be addressed: (1) how to optimize and formulate $f(\cdot; \boldsymbol{\phi})$, such that the graphs patched by $f(\cdot; \boldsymbol{\phi})$ enhance the performance of $f_g(\cdot; \boldsymbol{\theta}^*)$ over low-degree nodes; and (2) how to derive a unified learning scheme that allows $f(\cdot; \boldsymbol{\phi})$ to not only improve low-degree nodes but also maintain the GNN's original superiority over high-degree nodes.

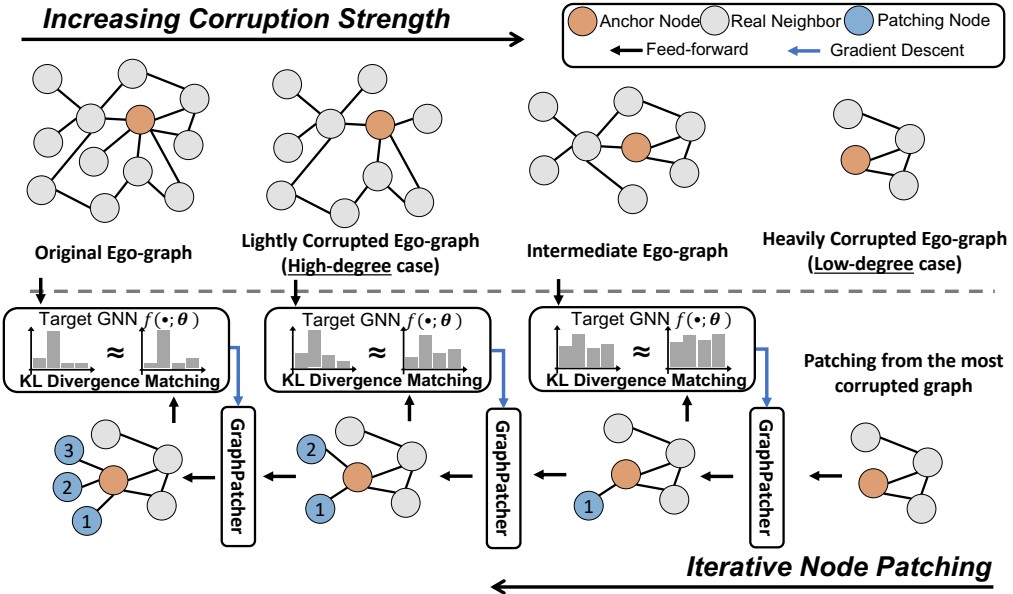

Figure 2: GRAPHPATCHER is presented ego-graphs corrupted by increasing strengths (i.e., the top half of the figure). From the most corrupted graph, it iteratively generates patching nodes to the anchor node, such that the target GNN behaves similarly given the currently patched graph or the corrupted graph next in the hierarchy (i.e., the bottom half of the figure).

## 3.2 The Proposed Framework: GRAPHPATCHER

Our proposed GRAPHPATCHER is a test-time augmentation framework for GNNs to mitigate their degree biases. As shown in Figure 2, GRAPHPATCHER is presented a sequence of ego-graphs corrupted by increasing strengths. Starting from the most corrupted graphs, GRAPHPATCHER iteratively generates patching nodes to augment the anchor nodes. Compared with the corrupted graphs next in the hierarchy, the patched graphs should allow the target GNN to deliver similar outputs. Through this scheme, GRAPHPATCHER not only learns how to patch low-degree nodes while preserving the superior performance over high-degree nodes.

### 3.2.1 Patching Ego-graphs via Prediction Reconstruction

In order to patch low-degree nodes, a straightforward approach is to corrupt high-degree nodes into low-degree nodes, and allowing the learning model to patch the corrupted nodes to restore their original properties [25, 13]. However, patching low-degree nodes not only affects their own representations but also those of their neighbors due to the message-passing mechanism of GNNs as well as the non-i.i.d. property of nodes in a graph. Besides, modeling over the entire graphs requires the learning model to consider all potential circumstances, whose overheads grow quadratically w.r.t. the number of nodes. Consequently, it becomes challenging to simultaneously determine both features and neighbors of the patching nodes given the entire graph.

To reduce the complexity of the optimization process, instead of working over the entire graph, we conduct node patching over ego-graphs and regard each ego-graph as an i.i.d. sample of the anchor node [56, 19]. For each node $v_i$, we have $f_g(G; \boldsymbol{\theta})[v_i] = f_g(\mathcal{G}(v_i); \boldsymbol{\theta})[v_i]$ if $k$ equals to the number of layers in $f_g(\cdot; \boldsymbol{\theta})$. To further simplify the optimization process, we directly wire the generated virtual nodes to the anchor node (i.e., the generated virtual nodes are the first-order neighbors of the anchor node). This implementation is simple yet effective, because we no longer consider the location to place the patching node: any modification that affects the latent representation of the anchor node can be achieved by patching nodes (with different features) directly to the anchor nodes.

We start explaining GRAPHPATCHER by the most basic case where we only conduct node patching once. Specifically, given the a trained GNN $f_g(\cdot; \boldsymbol{\theta}^*)$, an anchor node $v_i$, and a corruption function $\mathcal{T}(\cdot; t)$ with strength $t$ (i.e., first-order neighbor dropping with probability $t$ to simulate a low-degree

scenario), that is, $\mathcal{G}'(v_i) = (V'(v_i), E'(v_i)) = \mathcal{T}(\mathcal{G}(v_i), t)$. GRAPHPATCHER $f(\cdot; \phi)$ takes the corrupted ego-graph $\mathcal{G}'(v_i)$ as input and outputs the augmented ego-graph $\hat{\mathcal{G}}(v_i)$ with a patching node $v_p$ and its feature $\mathbf{x}_p$, which is directly connected to $v_i$. That is,

$$\hat{\mathcal{G}}(v_i) = f(\mathcal{G}'(v_i); \phi), \quad \text{where} \quad \hat{V} = V'(v_i) \cup \{v_p\}, \quad \hat{E} = E'(v_i) \cup \{e_{(i,p)}\}, \tag{2}$$

where $e_{(i,p)}$ refers to the edge connecting $v_i$ and $v_p$ and $V'(v_i)$ and $E'(v_i)$ refer to the nodes and edges in $\mathcal{G}'(v_i)$, respectively. To optimize $f(\cdot : \phi)$ such that $f_g(\cdot; \boldsymbol{\theta}^*)$ gives similar predictions to $\hat{\mathcal{G}}'(v_i)$ and $\mathcal{G}(v_i)$, we minimize the Kullback–Leibler divergence between the frozen GNN's predictions on these two ego-graphs, which is defined as:

$$\arg\min_{\phi} \sum_{v_i \in V_{\mathrm{tr}}} \text{KL-Div}\Big( f_g\big(\mathcal{G}(v_i); \boldsymbol{\theta}^*\big)[v_i], f_g\big(f(\mathcal{G}'(v_i); \phi); \boldsymbol{\theta}^*\big)[v_i]\Big), \tag{3}$$

where $\text{KL-Div}(\mathbf{y}_1, \mathbf{y}_2) = (\mathbf{y}_1 + \epsilon) \cdot \big( \log(\mathbf{y}_2 + \epsilon) - \log(\mathbf{y}_1 + \epsilon)\big)$[1] with $\epsilon > 0$ and $V_{\mathrm{tr}}$ refers to the set of anchor nodes for training. Intuitively, the reconstruction process above enforces GRAPHPATCHER to remedy the corrupted neighborhood caused by $\mathcal{T}(\cdot; t)$ via adding a patching node directly to the anchor node. It is philosophically similar to the existing works (e.g., TuneUp [13] and Tail-GNN [25]), where models gain better generalization over low-degree nodes via the corrupted high-degree nodes. Empirically, we observe that this branch of approaches can effectively enhance performance over low-degree nodes. Though promising, according to our empirical studies, it falls short on the high-degree node that original GNNs perform well at. This phenomenon may be attributed to the unintentially created out-of-distribution scenario [42], wherein models primarily encounter nodes with low degrees during the training. Consequently, the performance of GNNs, which is typically proficient with high-degree nodes, is adversely affected and downgraded.

### 3.2.2 Iterative Patching to Mitigate Degree Bias

In this work, we emphasize that: *mitigating degree bias should not focus specifically on the low-degree nodes: trading off performance on high-degree nodes for that on low-degree nodes simply creates a new bias towards high-degree nodes*. Therefore, besides enhancing the performance on low-degree nodes, maintaining GNN's original superiority on high-degree nodes is equally critical. This behavior is empirically desirable because practitioners can universally apply GRAPHPATCHER to all nodes without, like previous works do, manually discovering the degree threshold that differentiates the low- and high-degree nodes. Furthermore, the fact that these frameworks are applicable only to low-degree nodes indicates a lack of robustness: further remedying a neighborhood that is informative enough to deliver a good classification result should not jeopardize the performance.

To mitigate the degree bias, we propose a novel training scheme for GRAPHPATCHER such that it observes both low- and high-degree nodes simultaneously during the optimization. Specifically, given a node $v_i$, we firstly create a sequence of $M$ corrupted ego-graphs of $v_i$, denoted as $\mathcal{S}(v_i) = [\mathcal{G}'(v_i)_m = \mathcal{T}(\mathcal{G}(v_i), t_m)]_{m=1}^{M}$, with decreasing corruption strength (i.e., $\forall\ m, n \in \{1, \ldots, M\}$, $t_m > t_n$ if $m < n$). Instead of the one-step patching to match the prediction on the original ego-graph as described in Section 3.2.1, GRAPHPATCHER traverses $\mathcal{S}(v_i)$ and recursively patches the corrupted ego-graph to match the target GNN's prediction on the ego-graph next in the sequence. As also illustrated in Figure 2, this optimization process is formulated as:

$$\arg\min_{\phi} \sum_{v_i \in V_{\mathrm{tr}}} \sum_{m=1}^{M-1} \text{KL-Div}\Big( f_g\big(\mathcal{G}'(v_i)_{m+1}; \boldsymbol{\theta}^*\big)[v_i], f_g(\hat{\mathcal{G}}(v_i)_m; \boldsymbol{\theta}^*)[v_i]\Big), \tag{4}$$

$$\text{s.t. } \hat{\mathcal{G}}(v_i)_m = f(\hat{\mathcal{G}}(v_i)_{m-1}; \phi),$$

where $\hat{\mathcal{G}}(v_i)_m = (\hat{V}_m, \hat{E}_m)$ with $\hat{V}_m = V'_1(v_i) \cup \{v_p\}_{p=1}^{m}$, $\hat{E}_m = E'_1(v_i) \cup \{e_{(i,p)}\}_{p=1}^{m}$, and $\hat{\mathcal{G}}(v_i)_0 = \mathcal{G}'(v_i)_1$.

The one-step patching described in Section 3.2.1 remedies low-degree anchor nodes directly to the distributions of high-degree nodes. During this process, the model does not observe distributions of high-degree nodes and hence delivers sub-optimal performance. Therefore, we design GRAPH-PATCHER to be an iterative multi-step framework. At each step, it takes the previously patched

---

[1]KL divergence used here is equal to the regularized cross-entropy. It is strongly convex and Lipschitz continuous due to the incorporation of $\epsilon$. These two properties are required for the derivation of Theorem 1.

ego-graph as input and further remedies the partially patched ego-graph to match the GNN's prediction on the ego-graph next in the sequence. This scheme enables GRAPHPATCHER to learn to patch low-degree nodes in early steps when the ego-graphs are heavily corrupted (e.g., low-degree case in Figure 2) and maintain the original performance in later steps when ego-graphs are lightly corrupted (e.g., high-degree case in Figure 2). Specifically, at the $m$-th patching step, the currently patched ego-graph $\hat{\mathcal{G}}(v_i)_m$ reflects the neighbor distribution of ego-graphs corrupted by a specific strength of $t_{m+1}$. GRAPHPATCHER takes $\hat{\mathcal{G}}(v_i)_m$ as input and further generates another patching node $v_{m+1}$ to approach the neighbor distribution of ego-graphs corrupted by a slightly weaker strength of $t_{m+2}$. This process iterates until GRAPHPATCHER traverses $\mathcal{S}(v_i)$. Intuitively, the incorporation of $v_{m+1}$ enriches the neighbor distribution by an amount of $t_{m+2} - t_{m+1}$ corruption strength. This optimization scheme allows GRAPHPATCHER to observe neighbor distributions with varying corruption strengths and makes our proposal applicable to both low- and high-degree nodes.

However, the target distribution at each step (i.e., $f_g\big(\mathcal{G}'(v_i)_{m+1}; \boldsymbol{\theta}\big)[v_i]$ in Equation (4)) is not deterministic due to the stochastic nature of the corruption function $\mathcal{T}$. Given an ego-graph $\mathcal{G}(v_i)$ and a corruption strength $t$, one can at most generate $\binom{|V_i|}{(1-t)|V_i|}$ different corrupted ego-graphs. With a large corruption strength (e.g., ego-graphs early in the sequence $\mathcal{S}(v_i)$), two corrupted ego-graphs generated by the same exact priors might exhibit completely different topologies. Such differences could bring high variance to the supervision signal and instability to the optimization process. To alleviate the issue above, at each step we sample $L$ ego-graphs with the same corruption strength and let GRAPHPATCHER approximate multiple predictions over them, formulated as:

$$\mathcal{L}_{\text{patch}} = \sum_{v_i \in V_{\text{tr}}} \sum_{m=1}^{M-1} \sum_{l=1}^{L} \text{KL-Div}\Big( f_g\big(\mathcal{G}'(v_i)_{m+1}^l; \boldsymbol{\theta}^*\big)[v_i], f_g\big(\hat{\mathcal{G}}(v_i)_m; \boldsymbol{\theta}^*\big)[v_i]\Big), \tag{5}$$

where $\hat{\mathcal{G}}(v_i)_m = f(\hat{\mathcal{G}}(v_i)_{m-1}; \boldsymbol{\phi})$ and $\mathcal{G}'(v_i)_{m+1}^l$ refers to one of the $L$ target corrupted ego-graphs that GRAPHPATCHER aims to approximate at the $m$-th step. This approach allows GRAPHPATCHER to patch the anchor node towards a well-approximated region where its high-degree counterparts should locate, instead of one point randomly sampled from this region.

With $M-1$ virtual nodes patched to the ego-graph, we further ask GRAPHPATCHER to generate a last patching node to $\hat{\mathcal{G}}(v_i)_{M-1}$ and enforce the resulted graph $\hat{\mathcal{G}}'(v_i)_M$ to match the GNN's prediction on the original ego-graph. The last patching node could be regarded as a slack variable to complement minor differences between the original and the least corrupted ego-graphs, formulated as:

$$\mathcal{L}_{\text{recon}} = \sum_{v_i \in V_{\text{tr}}} \text{KL-Div}\Big( f_g\big(\mathcal{G}(v_i); \boldsymbol{\theta}^*\big)[v_i], f_g\big(\hat{\mathcal{G}}(v_i)_M; \boldsymbol{\theta}^*\big)[v_i]\Big), \tag{6}$$

where $\hat{\mathcal{G}}(v_i)_M = f(\hat{\mathcal{G}}(v_i)_{M-1}; \boldsymbol{\phi})$. $\mathcal{L}_{\text{recon}}$ (Equation (6)) also prevents GRAPHPATCHER from overfitting to the low-degree nodes and enforces GRAPHPATCHER to maintain the target GNN's performance over high-degree nodes, since only marginal distribution modification should be expected with this last patching node. Hence, GRAPHPATCHER is optimized by a linear combination of the above two objectives (i.e., $\arg\min_{\boldsymbol{\phi}} \mathcal{L}_{\text{patch}} + \mathcal{L}_{\text{recon}}$).

### 3.2.3 Theoretical Analysis

As shown in Equation (5), one of the important factors that contribute to the success of GRAPHPATCHER is sampling multiple ego-graphs with the same corruption strength. The following theorem shows that the error is bounded w.r.t. the number of sampled ego-graphs $L$.

**Theorem 1.** *Assuming the parameters of* GRAPHPATCHER *are initialized from the set* $P_\beta = \{\boldsymbol{\phi} : ||\boldsymbol{\phi} - \mathcal{N}(\mathbf{0}_{|\boldsymbol{\phi}|}; \mathbf{1}_{|\boldsymbol{\phi}|})||_F < \beta\}$ *where* $\beta > 0$, *with probability at least* $1 - \delta$ *for all* $\boldsymbol{\phi} \in P_\beta$, *the error (i.e.,* $\mathbb{E}(\mathcal{L}_{patch}) - \mathcal{L}_{patch}$*) is bounded by* $\mathcal{O}(\beta\sqrt{\frac{|\boldsymbol{\phi}|}{L}} + \sqrt{\frac{\log(1/\beta)}{L}})$.

The proof of Theorem 1 is provided in Appendix C. From the above theorem, we note that without the sampling strategy (i.e., $L = 1$), the generalization error depends only on the number of parameters (i.e., $|\boldsymbol{\phi}|$) given the same objective function, which could lead to high variance to the supervision signal and instability to the optimization process. According to this theorem and our empirical observation, an affordable value of $L$ (e.g., $L = 10$) delivers stable results across datasets.

# 4 Experiments

## 4.1 Experimental Setting

**Datasets**. We conduct comprehensive experiments on seven real-world benchmark datasets that are broadly utilized by the graph community, including `Cora`, `Citeseer`, `Pubmed`, `Wiki.CS`, `Amazon-Photo`, `Coauthor-CS`, `ogbn-arxiv`, `Actor`, and `Chameleon` [44, 28, 12, 33]. This list of datasets covers graphs with distinctive characteristics (i.e., graphs with different domains and dimensions) to fully evaluate the effectiveness of GRAPHPATCHER. The detail of these datasets can be found in Appendix A.

**Baselines**. We compare GRAPHPATCHER with six state-of-the-art graph learning frameworks from three branches. The first branch specifically aims at enhancing the performance on low-degree nodes, including TAIL-GNN [25], COLBBREW [55], and TUNEUP [13]. The second branch consists of frameworks that focus on handling out-of-distribution scenarios, including EERM [42] and GTRANS [15]. We list this branch of frameworks as baselines because the sub-optimal performance of GNNs over low-degree nodes could be regarded as an out-of-distribution scenario. As GRAPH-PATCHER is a test-time augmentation framework, the last branch of baseline includes DROPEDGE, which is a data augmentation framework employed during training.

**Evaluation Protocol**. We evaluate all models using the node classification task [22, 38], quantified by the accuracy score. For datasets with publicly avaiable (i.e., `ogbn-arxiv`, `Cora`, `Citeseer`, and `Pubmed`), we employ their the provided splits for the model training and testing. Whereas for other datasets, we create a random 10%/10%/80% split for the training/validation/testing split, to simulate a semi-supervised learning setting. All reported performance is averaged over 10 independent runs with different random seeds. Both mean values and standard deviations for the performances of all models are reported. Besides mitigating the degree bias for supervised GNNs, GRAPHPATCHER is also applicable to self-supervised GNNs. To evaluate the model performance for them, we apply GRAPHPATCHER and TUNEUP to state-of-the-art self-supervised GNNs including DGI [39], GRACE [57], and PARETOGNN [20]. We only compare our proposal with TUNEUP since other frameworks require specific model architectures and hence do not apply to self-supervised GNNs.

**Hyper-parameters**. We use the optimal settings on all baselines given by the authors for the shared datasets and a simple two-layer GCN [22] as the backbone model architecture for all applicable baselines. Hyper-parameters we tune for GRAPHPATCHER include learning rate, hidden dimension, the augmentation strength at each step, and the total amount of patching steps with details described in Appendix B. Besides, all of our models are trained on a single RTX3090 with 24GB VRAM; additional hardware information can also be found in the appendix.

## 4.2 Performance Comparison with Baselines

We compare GRAPHPATCHER with six state-of-the-art frameworks that mitigate the degree bias problem and the performances of all models are shown in Table 1. Firstly we notice that the problem of degree bias is quite serious across datasets for GCN. The performances on low-degree nodes are ∼10% lower than those over high-degree nodes. Comparing GCN with COLBBREW, TAIL-GNN, and TUNEUP, we can observe that frameworks that focus specifically on low-degree nodes can usually enhance GNN's performance over the lower percentile (e.g., 1.2% accuracy gain on Cora by TUNEUP, 0.74% on Citeseer by COLBBREW, 1.38% on Pubmed by TAIL-GNN, etc.). However, these frameworks fall short on the high-degree nodes and sometimes perform worse than the vanilla GCN (e.g., -2.7% accuracy degradation on Cora by TAIL-GNN, -11.42% on Wiki.CS by COLBBREW, and -2.5% on Amazon Photo by TUNEUP). This phenomenon could result from that they unintentionally create an artificial out-of-distribution scenario, where they only observe low-degree nodes during the training, leading to downgraded performance for high-degree nodes that GNNs originally perform well at. Comparing GCN with GTRANS and EERM, we observe that they deliver similar performances as the vanilla GCN does, indicating that frameworks targeting out-of-distribution scenarios cannot mitigate degree biases. Comparing GRAPHPATCHER with all baselines, we notice that our proposed GRAPHPATCHER consistently improves the low-degree performance with an average improvement gain of 2.23 accuracy score. Besides, unlike other frameworks that have downgraded performance over high-degree nodes, GRAPHPATCHER can maintain GCN's original high-degree superiority, due to our iterative node patching. On average, GRAPHPATCHER improves GCN's overall performance by a 1.4 accuracy score across datasets.

Table 1: Performance (%) of all models over nodes with different degrees. Lower and upper percentile indicate the set of nodes whose degree is ranked in the lower and upper 33% population respectively. A two-layer GCN is used as the backbone model for all applicable baselines. **Bold** indicates the best performance and underline indicates the runner-up, with standard deviations as subscripts.

| Method | Cora | Citeseer | Pubmed | Wiki.CS | Am.Photo | Co.CS | Arxiv | Chameleon | Actor |
|---|---|---|---|---|---|---|---|---|---|
| ACCURACY ON LOW-DEGREE NODES (LOWER PERCENTILE) | | | | | | | | | |
| GCN | $73.27_{\pm0.01}$ | $64.86_{\pm0.92}$ | $76.88_{\pm0.40}$ | $72.98_{\pm0.50}$ | $75.59_{\pm0.43}$ | $84.59_{\pm0.45}$ | $63.15_{\pm0.13}$ | $54.05_{\pm0.18}$ | $27.30_{\pm0.52}$ |
| COLBBREW | $73.82_{\pm0.98}$ | $65.60_{\pm0.08}$ | $77.72_{\pm0.63}$ | $73.98_{\pm0.52}$ | $76.18_{\pm0.80}$ | $85.56_{\pm0.69}$ | $63.02_{\pm0.21}$ | $53.41_{\pm0.22}$ | $\underline{27.88}_{\pm0.13}$ |
| TAIL-GNN | $71.17_{\pm0.80}$ | $57.66_{\pm0.83}$ | $75.38_{\pm0.89}$ | $\mathbf{74.36}_{\pm0.18}$ | $77.22_{\pm0.94}$ | $85.13_{\pm0.60}$ | OOM | $53.48_{\pm0.04}$ | $27.80_{\pm0.62}$ |
| TUNEUP | $\underline{74.47}_{\pm0.34}$ | $65.17_{\pm0.22}$ | $77.18_{\pm0.39}$ | $72.60_{\pm0.75}$ | $\underline{76.08}_{\pm0.62}$ | $84.68_{\pm0.50}$ | $63.34_{\pm0.32}$ | $53.87_{\pm0.43}$ | $27.94_{\pm0.14}$ |
| EERM | $73.40_{\pm0.06}$ | $64.27_{\pm0.33}$ | $76.30_{\pm0.20}$ | $73.12_{\pm0.68}$ | $75.15_{\pm0.59}$ | $84.82_{\pm0.74}$ | $63.20_{\pm0.11}$ | $\underline{54.11}_{\pm0.32}$ | $27.48_{\pm0.39}$ |
| GTRANS | $73.16_{\pm0.66}$ | $64.95_{\pm0.83}$ | $77.05_{\pm1.00}$ | $72.15_{\pm0.50}$ | $75.55_{\pm0.55}$ | $84.74_{\pm0.06}$ | $62.88_{\pm0.14}$ | $54.29_{\pm0.14}$ | $27.53_{\pm0.21}$ |
| DROPEDGE | $73.57_{\pm0.97}$ | $65.47_{\pm0.27}$ | $75.68_{\pm0.82}$ | $73.94_{\pm0.20}$ | $76.49_{\pm0.03}$ | $84.31_{\pm0.33}$ | $61.33_{\pm0.33}$ | $54.12_{\pm0.41}$ | $27.39_{\pm0.24}$ |
| GRAPHPATCHER | $\mathbf{78.08}_{\pm0.06}$ | $\mathbf{67.27}_{\pm0.20}$ | $\mathbf{78.98}_{\pm0.21}$ | $\underline{74.04}_{\pm0.86}$ | $\mathbf{77.84}_{\pm0.36}$ | $\mathbf{86.76}_{\pm0.84}$ | $\mathbf{64.01}_{\pm0.12}$ | $\mathbf{54.48}_{\pm0.71}$ | $\mathbf{29.27}_{\pm0.57}$ |
| ACCURACY ON HIGH-DEGREE NODES (UPPER PERCENTILE) | | | | | | | | | |
| GCN | $86.83_{\pm0.17}$ | $\mathbf{77.25}_{\pm1.00}$ | $80.84_{\pm0.76}$ | $83.40_{\pm0.70}$ | $84.07_{\pm0.71}$ | $90.20_{\pm0.37}$ | $80.46_{\pm0.18}$ | $54.11_{\pm0.73}$ | $27.41_{\pm0.29}$ |
| COLBBREW | $84.80_{\pm0.04}$ | $75.33_{\pm0.84}$ | $78.66_{\pm0.38}$ | $71.98_{\pm0.95}$ | $77.07_{\pm0.14}$ | $82.16_{\pm0.39}$ | $70.57_{\pm0.36}$ | $53.72_{\pm0.48}$ | $26.67_{\pm0.29}$ |
| TAIL-GNN | $84.13_{\pm0.48}$ | $74.85_{\pm0.30}$ | $78.74_{\pm0.34}$ | $78.91_{\pm0.97}$ | $80.32_{\pm0.60}$ | $86.75_{\pm0.90}$ | OOM | $\mathbf{54.53}_{\pm0.12}$ | $27.13_{\pm0.44}$ |
| TUNEUP | $\underline{87.13}_{\pm0.67}$ | $76.95_{\pm0.63}$ | $81.74_{\pm0.49}$ | $83.11_{\pm0.53}$ | $81.57_{\pm0.07}$ | $\mathbf{90.65}_{\pm0.86}$ | $80.09_{\pm0.51}$ | $54.25_{\pm0.59}$ | $26.64_{\pm0.71}$ |
| EERM | $85.89_{\pm0.09}$ | $76.32_{\pm0.23}$ | $79.98_{\pm0.06}$ | $82.98_{\pm0.07}$ | $84.32_{\pm0.96}$ | $90.17_{\pm0.11}$ | $80.37_{\pm0.12}$ | $\underline{54.41}_{\pm0.71}$ | $\underline{27.39}_{\pm0.14}$ |
| GTRANS | $86.32_{\pm0.34}$ | $76.60_{\pm0.44}$ | $80.56_{\pm0.92}$ | $\mathbf{83.42}_{\pm0.04}$ | $83.95_{\pm0.99}$ | $89.99_{\pm0.10}$ | $\mathbf{80.77}_{\pm0.26}$ | $54.21_{\pm0.19}$ | $27.29_{\pm0.12}$ |
| DROPEDGE | $86.53_{\pm0.99}$ | $76.35_{\pm0.17}$ | $81.44_{\pm0.51}$ | $\underline{83.37}_{\pm0.43}$ | $\mathbf{84.97}_{\pm0.56}$ | $89.28_{\pm0.08}$ | $\underline{80.64}_{\pm0.36}$ | $54.17_{\pm0.11}$ | $27.38_{\pm0.21}$ |
| GRAPHPATCHER | $\mathbf{88.02}_{\pm0.11}$ | $76.65_{\pm0.18}$ | $\mathbf{83.83}_{\pm0.79}$ | $83.49_{\pm0.22}$ | $84.17_{\pm0.97}$ | $90.59_{\pm0.46}$ | $80.61_{\pm0.25}$ | $54.20_{\pm0.21}$ | $\mathbf{27.43}_{\pm0.62}$ |
| OVERALL PERFORMANCE | | | | | | | | | |
| GCN | $81.22_{\pm0.40}$ | $70.51_{\pm0.46}$ | $79.14_{\pm0.31}$ | $77.30_{\pm0.41}$ | $80.38_{\pm0.86}$ | $88.16_{\pm0.66}$ | $71.73_{\pm0.14}$ | $52.83_{\pm0.35}$ | $27.20_{\pm0.57}$ |
| COLBBREW | $80.70_{\pm0.86}$ | $70.10_{\pm0.55}$ | $78.66_{\pm0.93}$ | $73.82_{\pm0.69}$ | $78.24_{\pm0.62}$ | $85.80_{\pm0.79}$ | $63.55_{\pm0.48}$ | $52.12_{\pm0.53}$ | $26.75_{\pm0.32}$ |
| TAIL-GNN | $79.44_{\pm0.64}$ | $65.80_{\pm0.04}$ | $76.14_{\pm0.25}$ | $74.66_{\pm0.18}$ | $80.68_{\pm0.58}$ | $87.02_{\pm0.33}$ | OOM | $52.62_{\pm0.47}$ | $27.62_{\pm0.47}$ |
| TUNEUP | $82.11_{\pm0.39}$ | $70.92_{\pm0.02}$ | $79.91_{\pm0.26}$ | $76.93_{\pm0.81}$ | $79.74_{\pm0.28}$ | $88.46_{\pm0.97}$ | $71.51_{\pm0.30}$ | $52.89_{\pm0.41}$ | $27.32_{\pm0.64}$ |
| EERM | $81.47_{\pm0.19}$ | $70.08_{\pm0.19}$ | $78.65_{\pm0.43}$ | $77.29_{\pm0.96}$ | $79.79_{\pm0.61}$ | $88.07_{\pm0.30}$ | $71.70_{\pm0.18}$ | $\underline{52.93}_{\pm0.24}$ | $27.65_{\pm0.29}$ |
| GTRANS | $80.79_{\pm0.51}$ | $69.51_{\pm0.93}$ | $78.67_{\pm0.93}$ | $76.39_{\pm0.27}$ | $80.02_{\pm0.80}$ | $88.06_{\pm0.96}$ | $71.77_{\pm0.19}$ | $52.67_{\pm0.35}$ | $26.93_{\pm0.41}$ |
| DROPEDGE | $81.10_{\pm0.31}$ | $71.10_{\pm0.86}$ | $78.90_{\pm0.68}$ | $77.49_{\pm0.78}$ | $81.11_{\pm0.72}$ | $87.56_{\pm0.64}$ | $71.82_{\pm0.33}$ | $52.89_{\pm0.22}$ | $27.28_{\pm0.32}$ |
| GRAPHPATCHER | $\mathbf{84.17}_{\pm0.54}$ | $\mathbf{71.65}_{\pm0.05}$ | $\mathbf{81.13}_{\pm0.68}$ | $\mathbf{78.12}_{\pm0.57}$ | $\mathbf{81.23}_{\pm0.32}$ | $\mathbf{89.44}_{\pm0.79}$ | $\mathbf{72.31}_{\pm0.22}$ | $\mathbf{53.21}_{\pm0.39}$ | $\mathbf{28.34}_{\pm0.24}$ |

Table 2: Performance (%) of GRAPHPATCHER and TUNEUP for different GNN architectures.

| Method | Cora | Citeseer | Pubmed | Wiki.CS | Am.Photo | Co.CS | Arxiv |
|---|---|---|---|---|---|---|---|
| ACCURACY ON LOW-DEGREE NODES (LOWER PERCENTILE) | | | | | | | |
| GCN | $73.27_{\pm0.01}$ | $64.86_{\pm0.92}$ | $76.88_{\pm0.40}$ | $72.98_{\pm0.50}$ | $75.59_{\pm0.43}$ | $84.59_{\pm0.45}$ | $63.15_{\pm0.13}$ |
| +TUNEUP | $74.47_{\pm0.34}$ | $65.17_{\pm0.22}$ | $77.18_{\pm0.39}$ | $72.60_{\pm0.75}$ | $76.08_{\pm0.62}$ | $84.68_{\pm0.50}$ | $63.34_{\pm0.32}$ |
| +GRAPHPATCHER | $\mathbf{78.08}_{\pm0.06}$ | $\mathbf{67.27}_{\pm0.20}$ | $\mathbf{78.98}_{\pm0.21}$ | $\mathbf{74.04}_{\pm0.86}$ | $\mathbf{77.84}_{\pm0.36}$ | $\mathbf{86.76}_{\pm0.84}$ | $\mathbf{64.01}_{\pm0.12}$ |
| G-SAGE | $70.57_{\pm0.84}$ | $67.44_{\pm0.11}$ | $76.58_{\pm0.36}$ | $\mathbf{61.83}_{\pm0.89}$ | $76.32_{\pm0.33}$ | $74.53_{\pm0.69}$ | $61.64_{\pm0.62}$ |
| +TUNEUP | $71.47_{\pm0.11}$ | $67.44_{\pm0.21}$ | $77.78_{\pm0.92}$ | $58.46_{\pm0.77}$ | $\mathbf{78.48}_{\pm0.89}$ | $74.88_{\pm0.76}$ | $62.43_{\pm0.59}$ |
| +GRAPHPATCHER | $\mathbf{72.33}_{\pm0.21}$ | $\mathbf{67.89}_{\pm0.33}$ | $\mathbf{78.93}_{\pm0.05}$ | $61.46_{\pm0.40}$ | $78.11_{\pm0.13}$ | $\mathbf{75.14}_{\pm0.21}$ | $\mathbf{62.92}_{\pm0.28}$ |
| GAT | $73.27_{\pm0.51}$ | $69.07_{\pm0.11}$ | $72.37_{\pm0.27}$ | $73.72_{\pm0.64}$ | $79.66_{\pm0.58}$ | $86.93_{\pm0.50}$ | $63.54_{\pm0.20}$ |
| +TUNEUP | $76.58_{\pm0.07}$ | $66.67_{\pm0.33}$ | $72.07_{\pm0.81}$ | $72.08_{\pm0.20}$ | $\mathbf{81.08}_{\pm0.25}$ | $86.72_{\pm0.84}$ | $63.71_{\pm0.31}$ |
| +GRAPHPATCHER | $\mathbf{76.88}_{\pm0.32}$ | $\mathbf{70.87}_{\pm0.78}$ | $\mathbf{74.47}_{\pm0.63}$ | $\mathbf{74.26}_{\pm0.72}$ | $80.05_{\pm0.19}$ | $\mathbf{89.50}_{\pm0.93}$ | $\mathbf{64.12}_{\pm0.14}$ |
| ACCURACY ON HIGH-DEGREE NODES (UPPER PERCENTILE) | | | | | | | |
| GCN | $86.83_{\pm0.17}$ | $\mathbf{77.25}_{\pm1.00}$ | $80.84_{\pm0.76}$ | $83.40_{\pm0.70}$ | $84.07_{\pm0.71}$ | $90.20_{\pm0.37}$ | $80.46_{\pm0.18}$ |
| +TUNEUP | $87.13_{\pm0.67}$ | $76.95_{\pm0.63}$ | $81.74_{\pm0.49}$ | $83.11_{\pm0.53}$ | $81.57_{\pm0.07}$ | $90.65_{\pm0.86}$ | $80.09_{\pm0.51}$ |
| +GRAPHPATCHER | $\mathbf{88.02}_{\pm0.11}$ | $76.65_{\pm0.18}$ | $\mathbf{83.83}_{\pm0.79}$ | $\mathbf{83.49}_{\pm0.22}$ | $\mathbf{84.17}_{\pm0.97}$ | $\mathbf{90.59}_{\pm0.46}$ | $\mathbf{80.61}_{\pm0.25}$ |
| G-SAGE | $82.04_{\pm0.01}$ | $72.46_{\pm0.70}$ | $80.24_{\pm0.12}$ | $60.83_{\pm0.18}$ | $77.30_{\pm0.56}$ | $69.24_{\pm0.55}$ | $78.66_{\pm0.08}$ |
| +TUNEUP | $80.84_{\pm0.36}$ | $\mathbf{73.95}_{\pm0.36}$ | $81.14_{\pm0.41}$ | $60.90_{\pm0.05}$ | $\mathbf{79.36}_{\pm0.89}$ | $70.12_{\pm0.25}$ | $79.26_{\pm0.51}$ |
| +GRAPHPATCHER | $\mathbf{82.14}_{\pm0.48}$ | $73.22_{\pm0.25}$ | $\mathbf{81.66}_{\pm0.46}$ | $\mathbf{61.02}_{\pm0.44}$ | $78.57_{\pm0.14}$ | $\mathbf{70.53}_{\pm0.68}$ | $\mathbf{79.91}_{\pm0.31}$ |
| GAT | $85.33_{\pm0.36}$ | $76.65_{\pm0.80}$ | $81.14_{\pm0.20}$ | $82.21_{\pm0.44}$ | $87.84_{\pm0.23}$ | $91.33_{\pm0.81}$ | $81.37_{\pm0.16}$ |
| +TUNEUP | $86.23_{\pm0.47}$ | $\mathbf{76.65}_{\pm0.33}$ | $80.84_{\pm0.02}$ | $81.73_{\pm0.36}$ | $\mathbf{89.02}_{\pm0.78}$ | $\mathbf{92.00}_{\pm0.04}$ | $81.44_{\pm0.11}$ |
| +GRAPHPATCHER | $\mathbf{86.53}_{\pm0.37}$ | $76.35_{\pm0.39}$ | $\mathbf{81.14}_{\pm0.89}$ | $\mathbf{82.34}_{\pm0.08}$ | $87.84_{\pm0.56}$ | $91.61_{\pm0.20}$ | $\mathbf{81.49}_{\pm0.15}$ |

We further apply GRAPHPATCHER to other GNN architectures (i.e., GraphSAGE [9] and GAT [38]) and compare its performance to TUNEUP. We only compare with TUNEUP since other baselines explore specific model architectures that do not allow a different backbone. From Table 2, we can observe that the issue of degree bias still exists on GAT and GraphSAGE with a performance gap between low- and high-degree nodes around ∼10%. Both TUNEUP and GRAPHPATCHER can improve the performance over low-degree nodes. Specifically, TUNEUP on average improves 0.27 low-degree accuracy for GraphSAGE and 0.40 for GAT across datasets; whereas GRAPHPATCHER improves 1.13 for GraphSAGE and 1.66 for GAT, outperforming TUNEUP by a large margin.

### 4.3 Performance of GRAPHPATCHER for Self-supervised GNNs

To fully demonstrate the effectiveness of GRAPHPATCHER, we also apply our proposal to self-supervised GNNs, as shown in Table 3. We can observe that self-supervised learning can mitigate degree bias by itself, proved by smaller gaps between low- and high-degree nodes than those of semi-supervised GNNs. Combined with GRAPHPATCHER, the degree biases can be further without sacrificing GNN's original superiority over high-degree nodes. On average, GRAPHPATCHER can enhance the low-degree performance of these three self-supervised GNNs by 1.78, 0.74, and 1.36 accuracy scores respectively.

Table 3: Effectiveness for self-supervised GNNs.

| Method | Cora | Pubmed | Wiki.CS |
|---|---|---|---|
| LOW-DEGREE NODES (LOWER PERCENTILE) | | | |
| DGI | $78.47_{\pm0.37}$ | $75.63_{\pm0.82}$ | $75.86_{\pm0.61}$ |
| +GRAPHPATCHER | $79.95_{\pm0.53}$ | $78.04_{\pm0.97}$ | $77.31_{\pm0.91}$ |
| GRACE | $77.81_{\pm0.73}$ | $77.80_{\pm0.65}$ | $74.31_{\pm0.63}$ |
| +GRAPHPATCHER | $78.53_{\pm0.82}$ | $78.49_{\pm0.16}$ | $75.12_{\pm0.34}$ |
| PARETOGNN | $78.85_{\pm0.71}$ | $78.32_{\pm0.33}$ | $74.17_{\pm0.18}$ |
| +GRAPHPATCHER | $79.91_{\pm0.62}$ | $79.11_{\pm0.89}$ | $76.41_{\pm0.22}$ |
| HIGH-DEGREE NODES (UPPER PERCENTILE) | | | |
| DGI | $86.83_{\pm0.82}$ | $81.14_{\pm0.28}$ | $81.09_{\pm0.81}$ |
| +GRAPHPATCHER | $86.91_{\pm0.10}$ | $82.31_{\pm0.53}$ | $80.95_{\pm0.19}$ |
| GRACE | $85.03_{\pm0.05}$ | $78.74_{\pm0.84}$ | $83.91_{\pm0.56}$ |
| +GRAPHPATCHER | $85.12_{\pm0.25}$ | $79.58_{\pm0.31}$ | $84.12_{\pm0.22}$ |
| PARETOGNN | $87.03_{\pm0.84}$ | $80.89_{\pm0.84}$ | $81.57_{\pm0.84}$ |
| +GRAPHPATCHER | $87.32_{\pm0.27}$ | $80.55_{\pm0.32}$ | $81.78_{\pm0.51}$ |

### 4.4 Effectiveness of GRAPHPATCHER for Enhancing SoTA Method

We apply GRAPHPATCHER to GRAND [5], a strong GNN that utilizes a random propagation strategy to perform graph data augmentation and significantly improve the node classification performance. The performance improvement brought by GRAPHPATCHER is shown in Table 4. We observe that GRAPHPATCHER can still consistently improve the node classification for GRAND. Specifically, on low-degree nodes, GRAPHPATCHER can improve 1.40, 2.23, and 4.20 accuracy score on Cora, Citeseer, and Pubmed, respectively. Overall, GRAPHPATCHER further enhances the SoTA perfor-

Table 4: Effectiveness for SoTA.

| Method | Cora | Citeseer | Pubmed |
|---|---|---|---|
| LOW-DEGREE NODES (LOWER PERCENTILE) | | | |
| GRAND | $80.18_{\pm0.64}$ | $70.57_{\pm0.68}$ | $80.48_{\pm0.14}$ |
| +GRAPHPATCHER | $81.58_{\pm0.45}$ | $72.73_{\pm0.29}$ | $84.68_{\pm0.29}$ |
| HIGH-DEGREE NODES (UPPER PERCENTILE) | | | |
| GRAND | $88.32_{\pm0.75}$ | $79.64_{\pm0.86}$ | $83.53_{\pm0.52}$ |
| +GRAPHPATCHER | $88.92_{\pm0.18}$ | $79.54_{\pm0.13}$ | $84.43_{\pm0.21}$ |
| OVERALL PERFORMANCE | | | |
| GRAND | $85.22_{\pm0.80}$ | $74.90_{\pm0.77}$ | $82.30_{\pm0.41}$ |
| +GRAPHPATCHER | $85.90_{\pm0.44}$ | $76.10_{\pm0.38}$ | $84.20_{\pm0.26}$ |

mance on these three datasets, with an outstanding accuracy score of 85.90, 76.10, and 84.20. The significant gain from GRAPHPATCHER indicates that the effectiveness brought by the test-time augmentation is not overlapped with the data augmentation during the training.

### 4.5 Performance w.r.t. the Number of Patching Nodes

To investigate the necessity of patching multiple nodes, we conduct experiments over the number of patching nodes at the test time. As shown in Figure 3, we notice that the overall performance gradually increments as the number of patching nodes increases, demonstrating that multiple patching nodes are required to remedy the incomplete neighborhood of low-degree nodes. Besides, we discover that the performance of GRAPHPATCHER saturates with around four nodes patched, which aligns with our training procedure, where the length of the ego-graph sequence is at most five. Experiments concerning the number of patching nodes during the optimization and the number of sampled ego-graphs per corruption strength (i.e., $M$ and $L$ in Equation (5)) can be found in Appendix B.

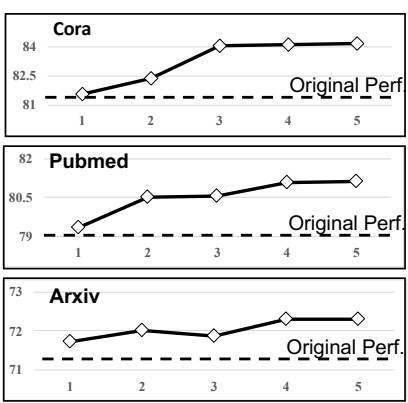

Figure 3: Overall perf. (y-axis) w.r.t. the number of patching nodes (x-axis).

## 5 Discussion w.r.t. Diffusion Models

Both diffusion models and GRAPHPATCHER conduct multiple corruptions to training samples with increasing strengths and generate examples in an iterative fashion. This scheme is conceptually inspired by heat diffusion from physics. However, the motivations behind them are different, where diffusion models focus on the generation quality (i.e., fidelity to the original data distribution) but ours

aims at the results brought by our generated nodes (i.e., the performance improvement). Specifically, diffusion models [10, 32] aim at learning the probability distribution of the data and accordingly generating examples following the learned distribution. Their goal is to generate samples that follow the original data distribution, agnostic of any other factor like the target GNN we have in our scenario. Whereas for GRAPHPATCHER, we aim at generating nodes to ego-nets such that the target GNN models deliver better predictions when the node degree is low. We mostly care about performance improvement and the generated node may be very different from the original nodes in the graph.

## 6  Discussion w.r.t. Generation Methods for Graph

Most graph generation frameworks (including those using diffusion models) explore iterative generation schemes to synthesize real graphs [58, 46, 6, 30, 16, 40]. They improve the generation quality and focus on applications such as molecule design, protein design, and program synthesis. Though GRAPHPATCHER also generates patching nodes for ego-graphs, ours is a different research direction than these methods. We do not focus on whether or not the generated patching nodes are faithful to the original data distribution, as long as the low-degree performance is enhanced and the high-degree performance is maintained. Another relevant work named GPT-GNN [14] explores an iterative node generation for pre-training, which also falls under the category of maintaining the original data distribution. In summary, GRAPHPATCHER is relevant to these frameworks in the sense that it generates nodes to add to ego-graphs. However, our proposal is motivated by a different reason and we aim at the performance improvement brought by generated nodes in downstream tasks.

## 7  Conclusion

We study the problem of degree bias underlying GNNs and accordingly propose a test-time augmentation framework, namely GRAPHPATCHER. GRAPHPATCHER iteratively patches ego-graphs with its generated virtual nodes to remedy the incomplete neighborhood. Through our designated optimization scheme, GRAPHPATCHER not only patches low-degree nodes but also maintains GNN's original superior performance over high-degree nodes. Comprehensive experiments are conducted over seven benchmark datasets and our proposal can consistently enhance GNN's overall performance by up to 3.6% and low-degree performance by up to 6.5%, outperforming all baselines by a large margin. Besides, GRAPHPATCHER can also mitigate the degree bias issue for self-supervised GNNs. When applied to graph learning methods with state-of-the-art performance (i.e., GRAND), GRAPHPATCHER can further improve the SoTA performance by a large margin, indicating that the effectiveness brought by the test-time augmentation is not overlapped with existing inductive biases.

## Limitation and Broader Impact

One limitation is the additional overhead entailed by generating ego-graphs. To address this limitation, we generate all ego-graphs before the optimization to avoid duplicated computations. This operation takes more hard-disk storage, which is relatively cheap compared with computational resources. Furthermore, we observe no ethical concern entailed by our proposal, but we note that both ethical or unethical applications based on graphs may benefit from the effectiveness of our work. Care should be taken to ensure socially positive and beneficial results of machine learning algorithms.

## Acknowledgement

We appreciate Shifu Hou from University of Notre Dame for valuable discussions and suggestions. We would also like to thank anonymous reviewers for their constructive suggestions and comments (i.e., experiments over heterophilic datasets, connections to diffusion models, and discussion w.r.t. iterative generation models for graphs). This work is partially supported by the NSF under grants IIS-2334193, IIS-2321504, IIS-2203262, IIS-2214376, IIS-2217239, OAC-2218762, CNS-2203261, and CMMI-2146076. Any opinions, findings, and conclusions or recommendations expressed in this material are those of the authors and do not necessarily reflect the views of any funding agencies.

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

## A  Dataset Description

We evaluate our proposed GRAPHPATCHER as well as other frameworks that mitigate the degree bias problem on seven real-worlds datasets spanning various fields such as citation network and merchandise network. Their statistics are shown in Table 5. For `Cora`, `Citeseer` and, `Pubmed`, we explore the community acknowledged public splits (i.e., fixed 20 nodes per class for training, 500 nodes for validation, and 1000 nodes for testing); whereas for `ogbn-arxiv`, we use the API from Open Graph Benchmark (OGB)[2] and explore the provided splits. For `Wiki.CS`, `Amazon-Photo`, and `Coauthor-CS`, we randomly select 10% nodes for training, another 10% for validation, and the remaining 80% for testing. We use the API from Deep Graph Library (DGL)[3] to load all datasets.

Table 5: Dataset Statistics.

| Dataset | # Nodes | # Edges | # Features | Avg. Degree | Split |
|---|---|---|---|---|---|
| Cora | 2,708 | 5,429 | 1,433 | 2.0 | Public Split |
| Citeseer | 3,327 | 4,732 | 3,703 | 1.4 | Public Split |
| Pubmed | 19,717 | 88,651 | 500 | 4.5 | Public Split |
| Wiki-CS | 11,701 | 216,123 | 300 | 18.5 | 10%/10%/80% |
| Amazon-Photo | 7,650 | 119,043 | 745 | 15.6 | 10%/10%/80% |
| Coauthor-CS | 18,333 | 81,894 | 6,805 | 4.5 | 10%/10%/80% |
| ogbn-arxiv | 169,343 | 1,166,243 | 128 | 6.9 | Public Split |

## B  GRAPHPATCHER Configuration and Experiment on Hyper-parameters

### B.1  GRAPHPATCHER Configuration

The architecture of GRAPHPATCHER consists of two parts; the first part is a 2-layer GCN encoder that takes an ego-graph as input and vectorizes its nodes and the second part is an MLP that takes the representation of the anchor node and outputs the generated feature for the virtual patching node.

To ensure the reproducibility, we also provide the detailed hyper-parameter configurations of GRAPHPATCHER for all datasets, as shown in Table 6. Besides, we use an early stopping strategy to decide the number of optimization steps, where the optimization stops if the validation loss stops decreasing for two consecutive steps.

Table 6: Hyper-parameters used for GRAPHPATCHER.

| Hyper-param. | Cora | Citeseer | Pubmed | Wiki.CS | Am.Photo | Co.CS | Arxiv |
|---|---|---|---|---|---|---|---|
| Augmentation strength | 0.3 | 0.3 | 0.3 | 0.3 | 0.3 | 0.3 | 0.1 |
| Patching step | 3 | 3 | 3 | 3 | 3 | 3 | 5 |
| # of sampled graphs | | | | 10 used for all datasets | | | |
| Batch size | 64 | 64 | 64 | 8 | 16 | 4 | 16 |
| Accumulation step | 16 | 16 | 16 | 32 | 16 | 16 | 64 |
| Learning rate | | | | 1e-4 used for all datasets | | | |
| Optimizer | | | AdamW with a weight decay of 1e-5 used for all datasets | | | | |

### B.2  Experiment on Hyper-parameters

The hyper-parameters we tune for GRAPHPATCHER include the number of patching nodes during the testing time, learning rate, hidden dimension, the augmentation strength at each step, and the total amount of patching steps. Experiments w.r.t. the number of patching nodes during the testing time has been showcased in Figure 3 and here we also append the results for the other four datasets, as shown in Figure 4. We observe similar trends as the aforementioned three datasets exhibit, where the

---

[2]https://ogb.stanford.edu
[3]https://www.dgl.ai

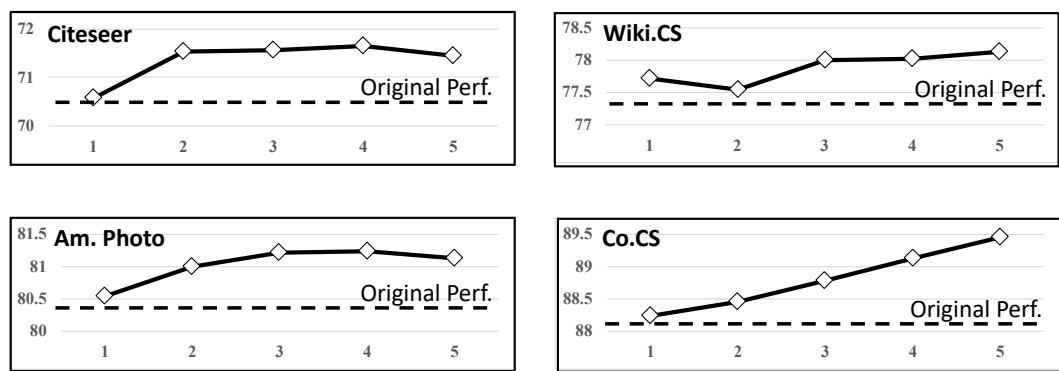

Figure 4: Overall perf. (y-axis) w.r.t. the number of patching nodes (x-axis).

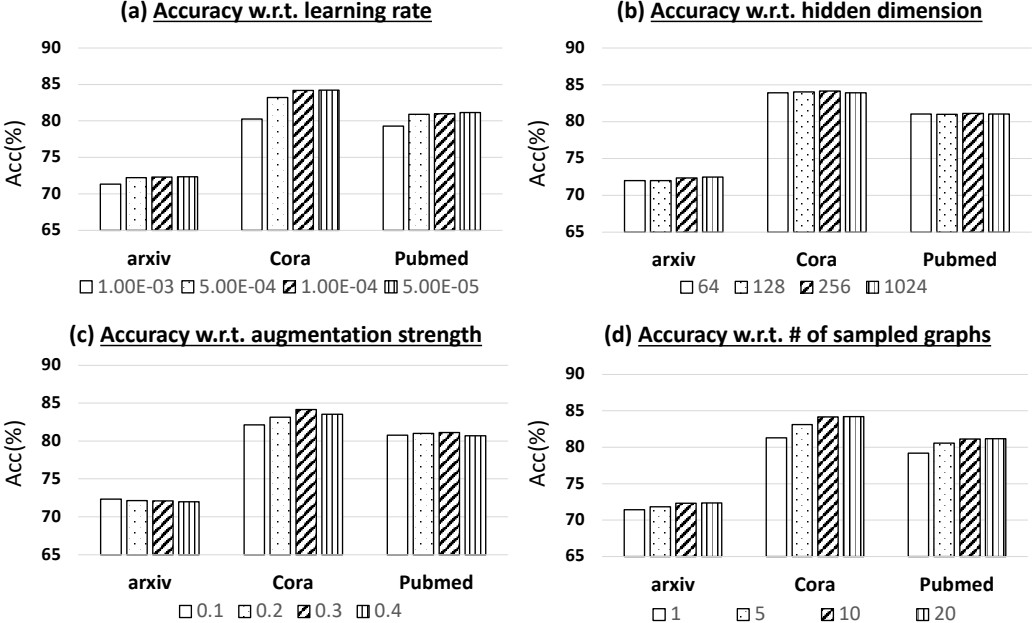

Figure 5: GRAPHPATCHER's sensitivity to different hyper-parameters.

performance of GRAPHPATCHER improves as the number of patching nodes increases and the gain saturates with 4 to 5 nodes patched.

We also conduct experiments w.r.t. learning rate, hidden dimension, the augmentation strength at each step, and the total amount of patching steps during the training. We tune the hidden dimension by conducting a grid search over common selections of [64, 128, 256, 1024] hidden units; we tune the learning rate similarly by searching over [1e-3, 5e-4, 1e-4, 5e-5]; and we tune the augmentation strength by searching over [0.1, 0.2, 0.3, 0.4].

The hidden dimension refers to the intermediate dimension of the 2-layer GCNs of GRAPHPATCHER. GRAPHPATCHER is constructed by a 2-layer GCN and features for virtual nodes are generated by a following multi-layer perceptron with the same hidden dimension. To reduce the search complexity, we explore an arithmetic sequence for the augmentation strength (i.e., the difference between any two consecutive strengths is the same) and set the total amount of patching steps during the training to $\lfloor \frac{1}{t} \rfloor$. For instance, an augmentation strength of 0.3 would lead to a 3-step training with augmentation strength of 0.3, 0.6, and 0.9 respectively. GRAPHPATCHER's sensitivity to these hyper-parameters is shown in Figure 5. Specifically, in Figure 5.(a) we can observe that across datasets, a large learning rate (i.e., 1e-3) leads to sub-optimal performance and GRAPHPATCHER achieves the best performance with a learning rate of 1e-4. We also investigate GRAPHPATCHER's sensitivity to the number of

hidden dimensions (i.e., the model size). In Figure 5.(b), we notice that for large graphs like `Arxiv`, the performance gradually increases as the model size enlarges. And for small and medium graphs like `Cora` and `Pubmed`, the performance saturates with a hidden dimension of 128. Besides, in Figure 5.(c) we study GRAPHPATCHER's performance w.r.t. the augmentation strength (which can also be interpreted as the number of patching steps as described previously). We can observe that, for small and medium graphs, strong augmentation strength leads to better performance, due to the sparsity of the graph structures. Whereas for large graphs, small augmentation strength delivers good performance. Furthermore, to prove the effectiveness of our proposed training scheme with multiple ego-graphs, we train GRAPHPATCHER with different numbers of sampled graphs (i.e., $L$ in Equation (5)), with the performance shown in Figure 5.(d). We can observe that without our proposed sampling strategy (i.e., the first column with $L = 1$), the performance of GRAPHPATCHER degrades significantly. As the number of sampled graphs gradually increases, the performance keeps improving and saturates with $L = 10$, empirically proving the effectiveness of the exploration of multiple ego-graphs for the same corruption strength.

### B.3 Hardware and Software Configuration

We conduct experiments on a server having one RTX3090 GPU with 24 GB VRAM. The CPU we have on the server is an AMD Ryzen 3990X with 128GB RAM. The software we use includes DGL 1.9.0 and PyTorch 1.11.0. As for the baseline models that we compare GRAPHPATCHER with, we explore the implementations provided by code repositories listed as follows:

- TAIL-GNN [25]: `https://github.com/shuaiOKshuai/Tail-GNN`.
- COLBBREW [55]: `https://github.com/amazon-science/gnn-tail-generalization`.
- EERM [42]: `https://github.com/qitianwu/GraphOOD-EERM`.
- GTRANS [15]L `https://github.com/ChandlerBang/GTrans`.
- DGI [39]: `https://github.com/dmlc/dgl/tree/master/examples/pytorch/dgi`.
- GRACE [57]: `https://github.com/dmlc/dgl/tree/master/examples/pytorch/grace`.
- PARETOGNN [20]: `https://github.com/jumxglhf/ParetoGNN`.

We sincerely appreciate the authors of these works for open-sourcing their valuable code and researchers at DGL for providing reliable implementations of these models. For TUNEUP [13], since the authors have not released the code yet, we manually implement it by ourselves, with a similar performance as reported in its original paper.

## C  Proof to Theorem 1

Here we re-state Theorem 1 before diving into its proof:

**Theorem 1.** *Assuming the parameters of* GRAPHPATCHER *are initialized from the set* $P_\beta = \{\phi : ||\phi - \mathcal{N}(\mathbf{0}_{|\phi|}; \mathbf{1}_{|\phi|})||_F < \beta\}$ *where* $\beta > 0$, *with probability at least* $1 - \delta$, *for all* $\phi \in P_\beta$, *the error bound (i.e.,* $\mathbb{E}(\mathcal{L}_{patch}) - \mathcal{L}_{patch}$*) is* $\mathcal{O}(\beta\sqrt{\frac{|\phi|}{L}} + \sqrt{\frac{\log(1/\beta)}{L}})$.

*Proof.* To prove Theorem 1, we need the following lemma, which has been broadly utilized in the literature of generalization error bound [26, 29].

**Lemma 1.** *Suppose a set* $P$ *of functions is* $(B, d)$-*Lipschitz parameterized for* $B > 0$ *and* $d \in \mathbb{N}$ *with input from a distribution* $D$ *and output in* $(0, 1)$. *There exist a constant* $c$ *such that for all* $n \in \mathbb{N}$, *for any* $\delta > 0$, *if* $S$ *is obtained by sampling* $n$ *times independently from* $D$, *with probability at least* $1 - \delta$, *for all* $B$ *and* $f \in P$, *we have:*

$$\mathbb{E}_{d \sim D}[f(d)] - \mathbb{E}_S[f] \leq c \cdot \left( B\sqrt{\frac{d}{n}} + \sqrt{\frac{\log(1/\delta)}{n}} \right). \tag{7}$$

In order to prove $\mathbb{E}(\mathcal{L}_{patch}) - \mathcal{L}_{patch}$ is $\mathcal{O}(\beta\sqrt{\frac{|\phi|}{L}} + \sqrt{\frac{\log(1/\beta)}{L}})$, we need to show that $\mathcal{L}_{patch}$ is Lipschitz continuous. $\mathcal{L}_{patch}$, as discussed in Section 3.2.1, is a regularized cross-entropy formulated

as $(\mathbf{y}_1 + \epsilon) \cdot \big( \log(\mathbf{y}_2 + \epsilon) - \log(\mathbf{y}_1 + \epsilon) \big)$. In this work, $\mathbf{y}_1$ and $\mathbf{y}_2$ refers to the prediction distribution (i.e., $0 < \mathbf{y}_1 < 1$) delivered by the GNN we aim at improving. Hence, we need to show that for given a specific $\mathbf{y}_1$, for any two $\mathbf{y}_2^a, \mathbf{y}_2^b \in \{\mathbf{y}_2' : 0 < \mathbf{y}_2' < 1\}$ and $K \in \mathbb{R}^+$, we have

$$\left\| (\mathbf{y}_1 + \epsilon) \cdot \log(\frac{\mathbf{y}_2^a + \epsilon}{\mathbf{y}_1 + \epsilon}) - (\mathbf{y}_1 + \epsilon) \cdot \log(\frac{\mathbf{y}_2^b + \epsilon}{\mathbf{y}_1 + \epsilon}) \right\|_F \le K \cdot \left\| \mathbf{y}_2^a - \mathbf{y}_2^b \right\|_F \tag{8}$$

$$\left\| (\mathbf{y}_1 + \epsilon) \cdot \big( \log(\frac{\mathbf{y}_2^a + \epsilon}{\mathbf{y}_1 + \epsilon}) - \log(\frac{\mathbf{y}_2^b + \epsilon}{\mathbf{y}_1 + \epsilon}) \big) \right\|_F \le K \cdot \left\| \mathbf{y}_2^a - \mathbf{y}_2^b \right\|_F \tag{9}$$

$$\left\| (\mathbf{y}_1 + \epsilon) \cdot \big( \log(\frac{\mathbf{y}_2^a + \epsilon}{\mathbf{y}_2^b + \epsilon}) \big) \right\|_F \le K \cdot \left\| \mathbf{y}_2^a - \mathbf{y}_2^b \right\|_F \tag{10}$$

Given the fact that $\log(\cdot)$ is strictly concave, Equation (10) holds and hence $\mathcal{L}_{\text{patch}}$ is Lipschitz continuous. We can then directly apply Lemma 1 to show that Theorem 1 holds. □

