# OpenReview forum: "GraphPatcher: Mitigating Degree Bias for Graph Neural Networks via Test-time Augmentation"
_NeurIPS.cc/2023/Conference — NeurIPS 2023 poster_

### Official Review · Reviewer_LJuX · 2023-06-28

**Soundness:** 2 fair
**Presentation:** 3 good
**Contribution:** 3 good
**Rating:** 5
**Confidence:** 3

**Summary:**

This paper addresses the issue of degree bias in node classification, which refers to the poorer performance of Graph Neural Network (GNN) models on nodes with lower degrees compared to the average level. While previous works have attempted to mitigate this bias, they often trade off the performance on higher-degree nodes for improvements on low-degree nodes. In light of this, the authors propose GraphPatcher, a method that introduces virtual nodes during the test phase to tackle this problem. To make the approach computationally feasible, the authors limit the search space to 1-order neighbors and devise an iterative procedure to repair a series of corrupted ego graphs by adding one virtual node at each step. Additionally, the authors provide theoretical results that establish the relationship between the number of sampled ego graphs and the accuracy of the estimated patching loss. The proposed GraphPatcher is extensively evaluated through experiments, demonstrating its effectiveness in addressing the degree bias issue.

**Strengths:**

1. The paper is well-written, allowing readers to easily grasp the core ideas and delve into the authors' detailed explanations.

2. The motivation behind the proposed GraphPatcher is well-founded, as it intuitively aims to enhance overall performance rather than compromising it through trade-offs.

4. The empirical evaluations consistently demonstrate the effectiveness of GraphPatcher, with statistically significant improvements observed. Furthermore, the increasing number of patched virtual nodes correlates with increasingly noticeable enhancements, providing strong support for the rationale behind the iterative procedure designed for GraphPatcher.

**Weaknesses:**

1. While the theoretical analysis provided in the paper is logically sound, its relevance to the community's primary concern of reducing the generalization risk of the original GNN model is unclear. Consequently, the presented theoretical results may not be perceived as a significant contribution.

2. Although the experiments in the paper are well-designed, there is room for improvement in terms of comprehensiveness. For instance, the inclusion of more visually informative figures/tables showcasing the improvements across different degree levels would enhance the clarity of the results. Additionally, comparisons on heterophilic graphs would provide a more comprehensive evaluation. It is also important to consider evaluating the efficiency of GraphPatcher since it introduces additional computational overhead.

**Questions:**

When working with graph samplers that are necessary for training on large graphs, whether we need a larger/smaller L?

---

> ### Author Rebuttal · Authors · 2023-08-10
>
> Dear Reviewer LJuX,
>
> Thank you for your valuable and kind feedback. We sincerely appreciate your acknowledgment of the good writing quality, motivation, and performance of our proposed framework. Our detailed response to your concerns is listed as follows:
>
> **[Relevance of our theoretical analysis]** \
> We appreciate your thoughtful consideration and valuable feedback. We recognize that the theorem we propose in our manuscript might not be directly applicable to any GNN in general. However, it is important to note that this theorem remains distinct from other GNNs, as it substantiates the iterative node patching and reconstruction processes, both of which are specifically tailored to our model. The significance of Theorem 1 lies in its demonstration that the upper bound of error for GraphPatcher can be notably reduced through the utilization of multiple sampled ego-graphs (i.e.,  Equation 5). We believe it serves as an important theoretical motivation for our sampling strategy. Along with the promising empirical results we have shown in our experiments, it further strengthens the credibility and substantiates the merits of our proposed GraphPatcher model.
>
> **[Performance on heterophilic datasets]** \
> Please refer to G2 in our general response for details. In summary, GraphPatcher exhibits similar trends for heterophilic datasets as we observe for homogeneous graphs. GraphPatcher improves the low-degree performance by 0.44 and 1.97 accuracy score and the overall performance by 0.38 and 1.14 on Chameleon and Actor respectively. Updated experiments over heterophilic datasets are included in our provided one-page pdf. We will update the final version of our paper with these additional experiments.
>
>
> **[Performance w.r.t. different degree levels]** \
> Thanks for raising this point -- we agree this would help clarify further where performance improvements are derived.  We have taken the liberty of incorporating additional plots into the one-page PDF that was submitted for the global rebuttal. In these new visuals, we meticulously analyze the performance of GCN, TuneUp, and GraphPatcher across a spectrum of 10 distinct degree ranges, ranging from the 0th to the 10th percentile, all the way up to the 90th to 100th percentile. Furthermore, we also include comparative graphs depicting the relative enhancements achieved by GraphPatcher and TuneUp in contrast to the baseline performance of GCN. These supplementary graphs exhibit consistent trends, mirroring the observations we made in the original manuscript regarding the coarse degree levels. Notably, GraphPatcher consistently showcases improvements in low-degree performance while concurrently preserving or even elevating high-degree performance. We appreciate your point and are looking forward  to incorporating these enlightening visuals into the appendix of our manuscript should it meet the criteria for acceptance. Your consideration of these additions would be greatly appreciated.
>
>
> **[Selection of L on large datasets]** \
> In the appendix (i.e., Figure 5 in Section B.2), we have presented GraphPatcher's performance in relation to the number of sampled graphs (referred to as L). Our observations indicate that consistently across all datasets, an L value below 10 tends to impact the overall performance. Notably, the performance reaches a saturation point at L=10, beyond which further increments in L do not yield discernible improvements across all datasets. As a result, we are led to the conclusion that, particularly when dealing with sizable graphs (such as the ogbn-arxiv dataset examined in this study), employing 10 sampled graphs is likely to yield promising and sufficient outcomes.
>
> We sincerely appreciate your suggestions for our manuscript. We will **accordingly modify the manuscript to clarify your concerns** when the anonymous period ends and we hope we have satisfactorily answered your questions. If so, could you please consider increasing your rating? If you have any remaining doubts or concerns, please let us know, and we will happily respond. Thank you!
>
> Best regards, \
> GraphPatcher's authors

---

> > ### Comment · Reviewer_LJuX · 2023-08-13
> > **Discussions**
> >
> > Thanks for your response! Most of my concerns are resolved. I will raise my score.

---

> > > ### Author Response · Authors · 2023-08-13
> > > **Thanks for your response.**
> > >
> > > We are delighted that our rebuttal satisfactorily leverages your concerns. We appreciate your constructive comments and will accordingly refine our manuscript.

---

### Official Review · Reviewer_Cs3J · 2023-07-03

**Soundness:** 2 fair
**Presentation:** 3 good
**Contribution:** 2 fair
**Rating:** 6
**Confidence:** 3

**Summary:**

This paper addresses degree bias in node classification. The authors show that current methods suffer performance degradation for high-degree nodes. Thus, they freeze the original GNN and train GraphPatcher to enhance low-degree nodes with node patching in the testing stage. In the experiments, they demonstrate that their approach significantly improves performance without performance drop in high-degree nodes on various homophilous graphs.

**Strengths:**

- They exhibit that the existing approaches suffer performance drops in high-degree nodes.
- Their method improves performance without performance degradation in high-degree nodes.

**Weaknesses:**

- [W1] From my understanding, it seems difficult to train in a node-parallel manner with a single large graph, as the graphs used by each node differ, unlike vanilla GNN training. Then, I’m concerned that training GraphPatcher might take an excessively long time.
- [W2] The validation is only conducted on homophilous graphs.
- [W3] I found that there is a performance drop for high-degree nodes on CiteSeer in Table 1. Is there any reason for this phenomenon?

**Questions:**

- Following [W1], could you provide the comparison with baselines from the perspective of the training time (including vanilla GNN)? Also, it would be better to show memory usage in GPU when training GraphPatcher and baselines.
- Following [W2], could you provide a performance comparison on heterophilous graphs?

**Limitations:**

As the authors mentioned, the additional computational cost might be considerable.

---

> ### Author Rebuttal · Authors · 2023-08-09
>
> Dear Reviewer Cs3J,
>
> Thank you for your valuable and kind feedback. We sincerely appreciate your acknowledgment of the good motivation and performance of our proposed framework. Our detailed response to your concerns is listed as follows:
>
>
> **[Node-parallel manner/Training time/Memory usage]** \
> Please refer to G3 in our general response for details. In summary, the training for GraphPatcher is fast and brings little extra computational overhead if we pre-compute all ego-graphs beforehand. With sufficient CPU cores, the training would be as fast even if we compute all ego-graphs on the fly, due to the
>
> **[Experiments over heterophilic datasets]** \
> Please refer to G2 in our general response for details. In summary, GraphPatcher exhibits similar trends for heterophilic datasets as we observe for homogeneous graphs. GraphPatcher improves the low-degree performance by 0.44 and 1.97 accuracy score and the overall performance by 0.38 and 1.14 on Chameleon and Actor respectively. Updated experiments over heterophilic datasets are included in our provided one-page pdf. We will update the final version of our paper with these additional experiments.
>
>
> **[Performance drop for high-degree nodes on Citeseer]** \
> We appreciate your attention to our experiments. We believe that the performance drop in Table 1 is a dataset-specific phenomenon. For instance, as you noticed, all models in Table 1 show sub-optimal performance for high-degree nodes in Citeseer. Most of the frameworks in this table explore graph convolution schemes similar to GCN. However, when the backbone model used is GraphSage, as shown in Table 2, both TuneUp and GraphPatcher improve Citeseer’s high-degree performance by 1.49 and 0.76 accuracy scores respectively. From these two ablations, we can conclude that the performance of high-degree nodes on Citeseer is sensitive to the model architecture.
>
> Moreover, when applied to self-supervised learning-based frameworks (even with GCN as the backbone model that does not favor high-degree nodes on Citeseer), both TuneUp and GraphPatcher can improve the high-degree performance, as demonstrated in Table 3. Hence, we can also conclude that the supervision signal (e.g., self-supervised learning vs. supervised learning) will also impact the high-degree performance.
>
> Recently, [1] discovers that there exist subgroups of nodes within each graph where different GNNs might exhibit distinct results, depending on factors such as the model architecture, supervision signal, etc. This paper might be a good reference for explanations of dataset-specific or subgroup-specific abnormal behaviors shown by different GNNs, and subgroup-level understanding is still an actively underexplored direction.
>
> We sincerely appreciate your suggestions for our manuscript. We will **accordingly modify the manuscript to clarify your concerns** when the anonymous period ends and we hope we have satisfactorily answered your questions. If so, could you please consider increasing your rating? If you have any remaining doubts or concerns, please let us know, and we will happily respond. Thank you!
>
> Best regards, \
> GraphPatcher's authors
>
> [1] Mao, Haitao, Zhikai Chen, Wei Jin, Haoyu Han, Yao Ma, Tong Zhao, Neil Shah, and Jiliang Tang. "Demystifying Structural Disparity in Graph Neural Networks: Can One Size Fit All?." arXiv preprint arXiv:2306.01323 (2023).

---

> > ### Comment · Reviewer_Cs3J · 2023-08-12
> > **Change my score to weak accept**
> >
> > I appreciate your detailed response. Most of my concerns are resolved. Thus, I change my score to weak accept.

---

> > > ### Author Response · Authors · 2023-08-12
> > > **Thanks for your response.**
> > >
> > > We are delighted that our rebuttal satisfactorily leverages your concerns. We appreciate your kind responses and acknowledgement of our work.

---

### Official Review · Reviewer_ZZgb · 2023-07-04

**Soundness:** 3 good
**Presentation:** 4 excellent
**Contribution:** 3 good
**Rating:** 6
**Confidence:** 4

**Summary:**

This paper introduces GraphPatcher, a novel test-time augmentation framework designed to mitigate degree bias in graph neural networks (GNNs). Degree bias causes GNNs to perform well with high-degree nodes (rich neighbor information) and poorly with low-degree nodes. Current strategies tend to focus on low-degree nodes during training, which can compromise performance on high-degree nodes. To overcome this, GraphPatcher creates virtual nodes to progressively 'patch' low-degree nodes, enhancing performance without sacrificing the capabilities of GNNs on high-degree nodes. The framework is model-agnostic, applicable to self-supervised or supervised GNNs, and shows significant improvement in overall and low-degree node performance across multiple benchmark datasets.

**Strengths:**

- I like the reasoning of the paper, which makes the proposed method well motivated and natural.
- The proposed method is more "adaptive" to both low-degree and high-degree nodes, without special treatments that is only for low degree node. The iterative patching sounds like denoising diffusion models, which is interesting.
- There are enough details for reproducibility and the code is attached to the supplementary.


**Weaknesses:**

**Concern About Weak Model Performances**: Though the experiment results show the GraphPatcher sometimes is more accurate than the baselines, it is less convincing because the baselines does not look strong enough. For example, for citation networks like Cora, Citeseer and Pubmed, the SOTA accuracy are much better than the best performance shown in Table 1. I recommend the authors to check https://paperswithcode.com/area/graphs for SOTA performances.

It is very important because there are non-negligible accuracy gaps. For example, for Citeseer, there are a lot of models achieved more than 80% accuracy, but in Table 1, the best model achieved only around 71% accuracy. Same for Cora (90% vs 84%), Pubmed (91% vs 81%).

I feel it would be better that the authors can evaluate the GraphPatcher on some SOTA models and evaluate the effect (improvement) of the proposed graph patching methods, rather than demonstrating improvements over some well-established models that is proposed years ago. It is crucial to understand the real-world value of the proposed methods in this fast-moving community.

**Questions:**

**Some Connections to Iterative Graph Generation And Diffusion Models**: The proposed GraphPatcher is conceptually to iterative graph generation framework, only the training objective is different. The way KL-divergence is used is also similar to a lot of graph generation methods. It is recommended to consider the connection to the graph generation works to see what can be adopted. I feel an autoregressive model like autoregressive transformer or LSTM can be a good candidate for generating the sequence of patched node. The main idea is that it could be possible to know when to stop (predict stop token).

The recent works on graph generation using diffusion models could also be a very relevant. These are just some random thoughts, I will appreciate the thoughts from the authors on how this could relate to this work, and maybe adding some future directions into the discussion section if it is indeed relevant.

**Limitations:**

No limitations are left unaddressed.

---

> ### Author Rebuttal · Authors · 2023-08-09
>
> Dear Reviewer ZZgb,
>
> Thank you for your valuable and kind feedback. We sincerely appreciate your acknowledgment of the motivation and reproducibility of our proposed framework. Our detailed response to your concerns is listed as follows:
>
> **[Baseline not strong]** \
> The baselines we compare GraphPatcher with are SoTA methods recently published on top venues (e.g., GTrans on ICLR’23, EERM and ColdBrew on ICLR’22, Tail-GNN on KDD’21, etc). We compare GraphPatcher with other test-time augmentation methods or cold-start/low-degree methods and evaluate by the relative improvement.
>
> By strong baselines, we believe you mean strong backbone models. If this is the right interpretation, we have self-supervised methods that can perform better than early models like GCN. For these strong methods (i.e., Table 3), GraphPatcher can still consistently improve the performance, aligning with our observation in the main table.
>
> To further address your concern, we explore the top model on the leaderboard (i.e., GRAND [1]) and apply GraphPatcher to it. As shown in the table below, GraphPatcher still enhances performance by a significant margin, with 1.5, 1.8, and 4.2 low-degree accuracy gains. Overall, GraphPatcher combined with GRAND can reach 85.9, 76.1, and 84.2 accuracy scores. The current SoTA scores are 85.5, 75.4, and 83.8 and we further improve them by 0.4, 0.7, and 0.4 accuracy scores, making **GraphPatcher the top 1 model on the leaderboard**. Please note that the leaderboard you were referring to is not the public split leaderboard. The correct one that aligns with our setting is “Cora/Citeseer/Pubmed with Public Split: fixed 20 nodes per class”. **We will update the leaderboard with the numbers we provide in this discussion** after the anonymous period and publicize the code.
> ||Cora|Citeseer|Pubmed|
> |:-|:-:|:-:|:-:|
> |||Low-degree|||
> |GRAND|80.18±0.64|70.57±0.68|80.48±0.14|
> |+GraphPatcher|81.68±0.45|72.37±0.29|84.68±0.29|
> |||High-degree|||
> |GRAND|88.32±0.75|79.64±0.86|83.53±0.52|
> |+GraphPatcher|88.92±0.18|79.54±0.13|41.13±0.21|
> |||Overall|||
> |GRAND|85.22±0.80|74.90±0.77|82.30±0.41|
> |+GraphPatcher|85.90±0.44|76.10±0.38|84.20±0.26|
>
> [1] Wenzheng Feng et al. "Graph random neural networks for semi-supervised learning on graphs." NeurIPS 2020.
>
> **[Connection to Iterative Graph Generation and Diffusion Models]** \
> We agree that our proposed framework is relevant to diffusion models and iterative graph generation models. Most graph generation frameworks (including those using diffusion models) explore iterative generation schemes to synthesize real graphs [1-7]. They improve the generation quality and focus on applications such as molecule design, protein design, and program synthesis. Though GraphPatcher also generates patching nodes for ego-graphs, ours is a different research direction than these methods. We do not focus on whether or not the generated patching nodes are faithful to the original data distribution, as long as the low-degree performance is enhanced and the high-degree performance is maintained. Another relevant work named GPT-GNN [8] explores an iterative node generation for pre-training, which also falls under the category of maintaining the original data distribution. In summary, GraphPatcher is relevant to these frameworks in the sense that it generates nodes to add to ego-graphs. However, our proposal is motivated by a different reason and we aim at the performance improvement brought by generated nodes in downstream tasks. Hence it is also difficult for us to directly compare GraphPatcher with these iterative generation models due to their different purposes and motivations. As for other diffusion models, we have detailed discussions in G1 in the general response. They are conceptually similar to this section but start from a more general point.
>
> **[Discussion on Auto-regressive Architecture]** \
> The architecture of GraphPatcher is auto-regressive where the node to be generated depends on previous generations. For mechanisms that terminate the generation like a stop token you mentioned, we believe it might be difficult in our setting. Unlike language generation where a sentence stops at its period, our node patching does not have such a ground truth. Right now we simply decide the generation length by the validation accuracy. A suitable heuristic could be the amount of changes (as measured in cosine similarity or other metrics) after a node patching. If the node representation stops updating with an additional node, then that indicates GraphPatcher is satisfied and the generation process could stop. We have not tested this idea yet and it might be a good further exploration.
>
> We sincerely appreciate your suggestions. We will **accordingly modify the manuscript to clarify your concerns** when the anonymous period ends and we hope we have satisfactorily answered your questions. If so, could you please consider increasing your rating? If you have any remaining doubts or concerns, please let us know, and we will happily respond. Thank you!
>
> Best regards, \
> GraphPatcher's authors
>
> [1] Yanqiao Zhu, et al. "A survey on deep graph generation: Methods and applications." LOG’22
>
> [2] Jiaxuan You, et al.. "Graphrnn: Generating realistic graphs with deep auto-regressive models." ICML’18
>
> [3] Nikhil Goyal et al. "Graphgen: A scalable approach to domain-agnostic labeled graph generation."  WWW’22
>
> [4] Chenhao Niu, et al. "Permutation invariant graph generation via score-based generative modeling." AISTATS’22.
>
> [5] Jaehyeong Jo et al.. "Score-based generative modeling of graphs via the system of stochastic differential equations." ICML’22
>
> [6] Clement Vignac, et al. "Digress: Discrete denoising diffusion for graph generation." ICLR’23
>
> [7] Meng Liu, et al. "Graphebm: Molecular graph generation with energy-based models." EBM@ICLR’23
>
> [8] Ziniu Hu, et al. "Gpt-gnn: Generative pre-training of graph neural networks." KDD’20

---

> > ### Comment · Reviewer_ZZgb · 2023-08-15
> >
> > Thank the authors for addressing my concerns. It seems there was some misunderstanding in the experiment setups, and I agree that it may be difficult to have a well-defined stopping criteria in graph patching. I believe my score has already acknowledged the contributions of this work, hence I will keep my score.

---

> > > ### Author Response · Authors · 2023-08-15
> > > **Thanks for your reply.**
> > >
> > > We greatly appreciate your valuable insights and constructive feedback on our paper. It's truly encouraging to see that our response effectively addresses the concerns you raised. Thank you for recognizing the efforts we've put into our work.

---

### Official Review · Reviewer_xNZm · 2023-07-05

**Soundness:** 2 fair
**Presentation:** 3 good
**Contribution:** 2 fair
**Rating:** 5
**Confidence:** 4

**Summary:**

The paper proposes GRAPHPATCHER, a test-time augmentation framework for graphs, to mitigate the degree biases in Graph Neural Networks. GRAPHPATCHER adopts a corruption function with increasing strength to simulate low-degree ego-graphs from a high-degree one. From the most corrupted graph, it then iteratively generates virtual nodes to the anchor node, such that the frozen GNN model behaves similarly given the currently patched graph or the corrupted graph next in the hierarchy. The authors then examine the effectiveness on seven real-world benchmark datasets, and observe that GRAPHPATCHER enhances the low-degree performance up to 6.5%, when enhancing the overall performance by up to 3.6% at the same time.

**Strengths:**

1. The proposed GRAPHPATCHER avoids creating an artificial out-of-distribution scenario when focusing only on low-degree nodes, and hence avoids significantly sacrificing the performance on high-degree nodes.
2. The modelling and optimization are conducted through aligning the ego-graph sequence and its reconstruction. The test-time augmentation framework avoids changing model architectures and the expensive re-training cost. The structure is simple with good performance.


**Weaknesses:**

1. The way GRAPHPATCHER patches the corrupted ego-graph (adding one node and the corresponding edge at a time) might limit the expressive ability of the model, as it could be difficult for this strategy to reflect more complex structures between the anchor node's neighbors.
2. The continuous corruption and patching on ego-graph share some similarities to the diffusion model. It would be more insightful if the theory behind the methodology and the similarity/dissimilarity with the diffusion model are discussed.
3. In the experimental part, Tail-GNN performs worse than GCN in almost all cases. Even for the low-degree nodes, Tail-GNN cannot always beat GCN, which is unexpected since Tail-GNN is specially designed for low-degree nodes. If there is a problem with the baseline tuning, some conclusions cannot be well supported (e.g., Tail-GNN falls short on the high-degree nodes). This also affects the convincingness of the superiority of GRAPHPATHCER over baselines.
4. It is claimed that the chosen datasets cover graphs with distinctive characteristics, but all datasets used are homophilic ones. It is expected to have more experiments on heterophilic datasets. Since heterophilic graphs often have more complex topologies, they are more susceptible to node adding and dropping.

Other minor issues:
Typo in row 8: "orizinally".


**Questions:**

Please refer to the weaknesses part.

**Limitations:**

The paper has properly addressed the limitations of the proposed method. One limitation is the additional computational cost entailed by generating ego-graphs. The authors provide solutions to mitigate the problem.

---

> ### Author Rebuttal · Authors · 2023-08-09
>
> Dear Reviewer xNZm,
>
> Thank you for your valuable feedback. We sincerely appreciate your acknowledgment of the effectiveness of our proposed framework. Our detailed response to your concerns is listed as follows:
>
> **[Limited expressive ability]** \
> We agree with you on the fact that the way we patch nodes to ego-graphs will not enhance the expressive ability from the perspective of graph-level isomorphism (e.g., making two ego-graphs distinguishable). However, **this is not a problem for our method** because we are not focusing on graph-level distinguishability (e.g., patching nodes to distinguish two ego-graphs that are indistinguishable before). Instead, we care about enhancing node-level predictions over low-degree nodes, where nodes with the same class are mapped to the same class after the node patching. For node classification tasks, a **stronger expressive ability might not always lead to good performance**. For instance, GIN has a stronger expressive ability than GCN, but GCN consistently outperforms GIN on many node classification benchmarks [3]. Moreover, training-time augmentation frameworks that add additional edges/nodes to the graph might also impact the expressive ability of the learning model [1,2]. Furthermore, even for graph-level classification tasks where expressive ability is very important, a model with better expressive ability does not necessarily outperform models with lower expressive ability [3,4].
>
> [1] Zhao, Tong, Yozen Liu, Leonardo Neves, Oliver Woodford, Meng Jiang, and Neil Shah. "Data augmentation for graph neural networks." AAAI’21
>
> [2] Mehdi Azabou, Venkataramana Ganesh, Shantanu Thakoor, Chi-Heng Lin, Lakshmi Sathidevi, Ran Liu, Michal Valko, Petar Veličković, and Eva L Dyer. "Half-Hop: A graph upsampling approach for slowing down message passing." ICML’23
>
> [3] Dwivedi, Vijay Prakash, Chaitanya K. Joshi, Anh Tuan Luu, Thomas Laurent, Yoshua Bengio, and Xavier Bresson. "Benchmarking Graph Neural Networks." JMLR’23
>
> [4] Zhao, Lingxiao, Neil Shah, and Leman Akoglu. "A practical, progressively-expressive GNN." NeurIPS’22
>
> **[Similarity/Dissimilarity with the diffusion model]** \
> Please refer to G1 in the general response. We will update the final version of our paper by adding the discussion above and we sincerely appreciate your suggestions.
>
> **[Performance of Tail-GNN]** \
> Thanks for your attention to our experiments. We fully acknowledge the contributions brought by Tail-GNN as it is one of the earliest works that enhance low-degree performance by learning from corrupted high-degree nodes. When the evaluation setting aligns with Tail-GNN’s motivation, the low-degree performance indeed is improved by a lot. As shown in our results, Tail-GNN’s low-degree performances on dense datasets like Wiki. CS, Am. Photo and Co. CS are competitive (i.e., 1st on Wiki. CS, 2nd on Am. Photo, and 3rd on Co. CS) compared to GCN and other SoTA methods. However, Tail-GNN is well motivated by solving the cold start problem and it is evaluated in a setting specifically designed for low-degree nodes. Tail-GNN explores warm nodes only for training (i.e., all nodes with a degree higher than 5 across all datasets) and cold nodes only for testing (i.e., all the remaining nodes). This setup enables Tail-GNN to perform well on low-degree nodes over dense datasets like Wiki. CS, Co. Photo, and Co. CS.
>
> Unlike TailGNN, GraphPatcher explores a different research direction: while enhancing the low-degree node performance, GraphPatcher also aims at maintaining good performances over high-degree nodes. In order to evaluate this goal, we need to train and evaluate models by a mixture of low- and high-degree nodes. This setting might not favor Tail-GNN as it is different from the original setting that Tail-GNN is evaluated on. We followed the optimal hyper-parameters provided by the authors of Tail-GNN in the original paper and utilized the official GitHub repo to conduct the experiment. For Cora, Citeseer, and Pubmed, they are relatively sparse with average degrees lower than 5, which also does not favor Tail-GNN due to the lack of high-degree nodes for Tail-GNN to train from.
>
> **[Experiments over heterophilic datasets]** \
> Please refer to G2 in the general response. The updated experiments over heterophilic datasets are included in our provided one-page pdf. We will update the final version of our paper with these additional experiments.
>
> We sincerely appreciate your suggestions over our manuscript. We will **accordingly modify the manuscript to clarify your concerns** when the anonymous period ends and we hope we have satisfactorily answered your questions. If so, could you please consider increasing your rating? If you have any remaining doubts or concerns, please let us know, and we will happily respond. Thank you!
>
> Best regards, \
> GraphPatcher's authors

---

> > ### Comment · Reviewer_xNZm · 2023-08-16
> > **Response after rebuttal**
> >
> > Thank you for the rebuttal. It has addressed my concerns. I have raised my rating.

---

> > > ### Author Response · Authors · 2023-08-16
> > > **Thanks for your reply.**
> > >
> > > Your valuable insights and constructive feedback on our paper are deeply appreciated. It is motivating to observe that our response has effectively tackled your concerns. We are grateful for acknowledging the dedication we've invested in our research.

---

### Author Rebuttal · Authors · 2023-08-09

Dear ACs and reviewers,

We thank the reviewers for their feedback and constructive suggestions. We are pleased that most reviewers appreciated **the promising effectiveness and performance of our framework**, e.g.: “simple with good performance” (xNZm), “method improves performance without performance degradation” (Cs3J), and “statistically significant improvements observed” (LJuX). Moreover, most reviewers also acknowledged **the good motivations behind our framework**, said: “the proposed method well motivated and natural” (ZZgb), “exhibit that the existing approaches suffer performance drops in high-degree nodes” (Cs3J), and “motivation behind the proposed GraphPatcher is well-founded” (LJuX).

**[G1: Connection between diffusion models and GraphPatcher]** \
 At the same time, multiple reviewers asked about the connection between diffusion models and our GraphPatcher. We appreciate the insightful suggestions and agree that GraphPatcher is relevant to diffusion models. Both diffusion models and our proposal conduct multiple corruptions to the training samples with increasing strengths and generate examples in an iterative fashion. This scheme is conceptually inspired by heat diffusion from physics. However, the motivations behind them are different, where diffusion models focus on the generation quality (i.e., fidelity to the original data distribution) but ours aims at the results brought by our generated nodes (i.e., the performance improvement). Specifically, diffusion models aim at learning the probability distribution of the data and accordingly generating examples following the learned distribution. Their goal is to generate samples that follow the original data distribution (i.e., P(X)), agnostic of any other factor like the target GNN we have in our scenario. Whereas for GraphPatcher, we aim at generating nodes to ego-nets such that the target GNN models deliver better predictions (i.e., P(Y|X)) when the node degree is low. We mostly care about performance improvement and the generated node may be very different from the original nodes in the graph.

**[G2: Experiments on heterophilic datasets]** \
Besides, reviewers also worried about GraphPatcher’s effectiveness over heterophilic graphs. We agree that incorporating heterophilic datasets would help us better understandGraphPatcher’s behavior, and also benefit readers. We have accordingly added experiments on two heterophilic datasets (i.e., Chameleon and Actor), with results shown in the table from our provided one-page pdf.  One interesting behavior of these two heterophilic datasets is that even though the degree distribution is power-law, GNN’s performance doesn’t downgrade much on low-degree nodes compared with the performance on high-degree nodes. As shown in the table, GCN’s performances on low- and high-degree percentiles are pretty close. On these two heterophilic datasets, GraphPatcher universally improves GNN’s performance across all degrees. Specifically, GraphPatcher improves the low-degree performance by 0.44 and 1.97 accuracy score and the overall performance by 0.38 and 1.14 on Chameleon and Actor respectively.

**[G3: Efficiency and running time]** \
Some reviewers raised concerns about the efficiency and running time of our proposed GraphPatcher. While designing GraphPatcher, we paid attention to its scalability and potential applications to industrial pipelines, which is one of the reasons why we explore the ego-net design. In a single forward pass, we only generate a fixed number of ego-graphs (i.e., mostly 16 or 64 ego-graphs per batch across our experiments) for the back-propagation. In this setting with small batch sizes, GraphPatcher can already deliver good performance improvements to target GNN models with light GPU usage and little computational overhead compared with baselines, with the GPU usage and running time shown in the table below.
|||TuneUp|Tail-GNN|GraphPatcher|
|:-|:-:|:-:|:-:|:-:|
| | |Cora | | ||
| Training Time (s) | 2.3 |17.7|14.5|8.1|
| Train+Eval Time (s)| 2.3|17.7|14.5| 13.3 (34.8)|
|GPU Consumption (GB)| 1.4|2.0|1.6|3.1|
| | |Citeseer         | | ||
| Training Time (s) | 2.6 |9.7|35.4|8.3|
| Train+Eval Time (s)| 2.6|9.7|35.4|12.1 (32.3)|
|GPU Consumption (GB)| 1.4|1.5|3.8|2.2|
| | |Pubmed         | | ||
| Training Time (s) | 3.2 |17.3|47.6|9.1|
| Train+Eval Time (s)| 3.21|17.3|47.6|14.4 (36.9)|
|GPU Consumption (GB)| 1.6|1.7|1.9|3.4|

The training for GraphPatcher is fast and brings little extra computational overhead. When all the evaluation ego-graphs are pre-computed beforehand, the total running times for Cora, Citeseer, and Pubmed are under 15 seconds (i.e., 13.3, 12.1, and 14.4 seconds). However, as we mentioned in the limitation section, if evaluation ego-graphs are extracted on the fly, the running time will be jeopardized (i.e., 34.8, 32.3, and 36.9 for these three datasets). This issue can be easily resolved by using more CPU cores as this step is embarrassingly parallelizable (currently we only use 24 cores). With sufficient CPU cores, the batch loading time would be close to the time needed to extract 1 ego-net, which only brings neglectable overhead compared with loading them from the disk.

We will **accordingly modify the manuscript to clarify your concerns** when the anonymous period ends, and we hope we have satisfactorily answered your questions. If so, could you please consider increasing your rating? If you have any remaining doubts or concerns, please let us know, and we will happily respond. Thank you!

Best regards,\
GraphPatcher's authors

---

### Decision · Program_Chairs · 2023-09-21

**Decision:**

Accept (poster)

**Comment:**

This paper proposes an interesting and novel technique for ameliorating the degree bias issue in GNNs, leading to measurable improvements in performance. While the reviewers initially pointed out several limitations in the work as presented, the Authors have put together a very convincing rebuttal, which led to a unanimously positive assessment of the paper. I congratulate the authors for their efforts, and recommend this paper for acceptance without reserve.